

# Emissions of trace gases from Australian temperate forest fires: emission factors and dependence on modified combustion efficiency

Elise-Andrée Guérette[1], Clare Paton-Walsh[1], Maximilien Desservettaz[1], Thomas E.L Smith[2],
Liubov Volkova[3], Christopher J. Weston[3], and C.P. (Mick) Meyer[4]

[1]Centre for Atmospheric Chemistry, School of Chemistry, University of Wollongong, Wollongong, NSW, Australia
[2]Department of Geography, King's College London, London, UK
[3]School of Ecosystem and Forest Sciences, the University of Melbourne, Creswick, VIC, Australia
[4]CSIRO Oceans and Atmosphere Flagship, Aspendale, VIC, Australia

*Correspondence to:* E-A. Guérette (eag873@uowmail.edu.au)

**Abstract.** We characterised trace gas emissions from Australian temperate forest fires through a mixture of in situ open-path FTIR measurements spectroscopy and selective ion flow tube mass spectrometry (SIFT-MS) and White cell FTIR spectroscopy of grab samples. We report emission factors for a total of 25 trace gas species measured in smoke from nine prescribed fires. We find significant dependence on modified combustion efficiency (MCE) for some species, although regional differences

indicate that the use of MCE as a proxy may be limited. We also find that the fire-integrated MCE values derived from our in situ on-the-ground open-path measurements are not significantly different from those reported for airborne measurements of smoke from fires in the same ecosystem. We then compare our average emission factors to those measured for fires in North American temperate ecosystems and for fires in Australian savanna and find that, although emission factors of some species agree within 20%, others differ by a factor of 2 or more. This indicates that the use of ecosystem-specific emission factors is

warranted for applications involving emissions from Australian forest fires.

## 1   Introduction

Biomass burning emits a wide range of trace species, including greenhouse gases, particulate matter and volatile organic compounds (VOCs). Globally, fires are the second largest source of VOCs, with emissions estimated at 400 Tg yr$^{-1}$ on average (Yokelson et al., 2008; Akagi et al., 2011). Fires are also the main driver of inter-annual variability for species such as

carbon monoxide and aerosol (Edwards et al., 2004, 2006; Voulgarakis et al., 2015).

Australia emits 7-8 % of global annual biomass burning carbon emissions (Ito and Penner, 2004; van der Werf et al., 2010). At a national level, average gross annual emissions of total carbon from fires (127 Tg C yr$^{-1}$) actually exceed those from burning fossil fuels (95 Tg C yr$^{-1}$) (Haverd et al., 2013). While net emissions of carbon from fires are lower due to rapid regrowth, volatile organic species emitted by those fires are not subject to uptake by the regenerating vegetation and can

therefore be considered net emissions.

The mix of VOCs emitted during biomass burning may be ecosystem-specific, especially for VOCs that are associated with biogenic processes (as opposed to combustion processes) and that are distilled from the vegetation in the early stages of the



fire (Ciccioli et al., 2014). Species such as methanol, acetic acid, acetaldehyde, acetone and monoterpenes have been detected from heated *Eucalyptus* leaves in laboratory experiments, with differences observed between fresh leaves and senescent leaves (Greenberg et al., 2006; Maleknia et al., 2007, 2009; Possell and Bell, 2013). Other factors that impact smoke composition include fuel composition (e.g. nitrogen content, Coggon et al., 2016) and fire behaviour (e.g. Wooster et al., 2011). Changes in

fire behaviour can be reflected in the combustion efficiency of the fire, i.e. in the proportion of total carbon that is emitted as $CO_2$. A useful proxy for combustion efficiency is modified combustion efficiency (MCE), which is defined as the ratio of $CO_2$ released to the sum of CO and $CO_2$ (Hao and Ward, 1993; Yokelson et al., 1996). Emission factors of several trace gases have been found to correlate to MCE in a number of ecosystems (e.g. Akagi et al., 2013; Burling et al., 2011; Meyer et al., 2012).

The composition of fresh smoke matters as it affects plume chemistry as the smoke ages, contributing to varying rates

of ozone and aerosol formation (Yokelson et al., 2009; Akagi et al., 2012; Alvarado et al., 2015) and elevated ozone and particulates downwind of the fires (Pfister et al., 2008; Yan et al., 2008).

Most of the area burnt in Australia annually is in the semi-arid and tropical savannas in the north of the country (Russell-Smith et al., 2007), but large bushfires also occur regularly in the temperate forests that cover extensive areas of the south-east of Australia (Cai et al., 2009). These fires can be intense enough to create pyro-convective lofting and inject smoke at high

altitudes (Fromm et al., 2006; Dirksen et al., 2009; Guan et al., 2010) and are expected to become more frequent under a changing climate (Bradstock et al., 2009; Cai et al., 2009; Keywood et al., 2013; King et al., 2013). There has been growing interest in characterising the composition of smoke from Australian temperate forest fires in recent years, mostly arising from increased awareness of the significant impacts of bushfire smoke on regional air quality (Reisen et al., 2011, 2013; Price et al., 2012; Keywood et al., 2015; Rea et al., 2016) and its associated repercussions on human health (Reisen and Brown, 2006;

Johnston et al., 2012, 2014; Reisen et al., 2015; Reid et al., 2016), coincident with a mandate for state agencies to increase prescribed burning in the wake of the catastrophic 2009 forest fires in Victoria (Teague et al., 2010). Prescribed burning is widely used in Australia as a means of reducing bushfire risk (Boer et al., 2009); however, these low to moderate intensity fires often take place close to population centres, under weather conditions that are conducive to pollution build up, sometimes on a regional scale (e.g., Williamson et al., 2016, Fig. 2), with potential health impacts on nearby population (Haikerwal et al.,

25    2015).

Most of what is known about the VOC emissions from Australian temperate forest fires to date comes from opportunistic measurements of bushfire plumes impacting measurement sites such as the University of Wollongong (Paton-Walsh et al., 2005, 2008; Rea et al., 2016) or the Cape Grim Baseline Air Pollution Station (Lawson et al., 2015) or captured from space using satellite sensors (Young and Paton-Walsh, 2011; Glatthor et al., 2013). Dedicated field and laboratory measurement campaigns

have mostly focused on greenhouse gases (Hurst et al., 1996; Volkova et al., 2014; Possell et al., 2015; Surawski et al., 2015) and only one study reports emission factors that can be deemed representative of whole fires (Hurst et al., 1996).

Volkova et al. (2014) reported emission factors for $CO_2$, CO, $CH_4$ and $N_2O$ separately for burning fine fuels and logs from measurements made on the ground at prescribed fires in the State of Victoria. Surawski et al. (2015) measured emissions of $CO_2$, CO, $CH_4$ and $N_2O$ from fine *Eucalyptus* litter fuels in a combustion wind tunnel and found that emissions from these

fuels vary depending on the mode of fire spread and on the phase of combustion. Possell et al. (2015) reported emission factors





for $CO_2$ and CO for several fuel classes combusted in a mass-loss calorimeter and estimated the total fraction of fuel carbon that would be emitted as $CH_4$, particulates and non-methane hydrocarbons using a carbon mass balance approach. The only whole fire emission factors available are those from Hurst et al. (1996), who sampled smoke plumes from fires in the greater Sydney region from an aircraft and reported emission factors for $CO_2$, CO and $CH_4$.

This paper presents results from a dedicated ground measurement program that sampled smoke at several prescribed fires organised by the New South Wales National Parks and Wildlife Service in the greater Sydney area and by the Department of Environment, Land, Water and Planning in the State of Victoria. Measurements made at a subset of these fires were presented in Paton-Walsh et al. (2014) along with a detailed description of the open-path Fourier Transform Infrared system (OP-FTIR) and a discussion of the uncertainties associated with deriving emission factors using this technique. Here, we present emission

factors for 15 additional VOC species, measured by selected ion flow tube mass spectrometry (SIFT-MS) from grab samples collected at prescribed fires in NSW, as well as additional OP-FTIR results from fires in the State of Victoria. We then investigate the dependence of the measured emission factors on MCE, using all the data collected to date. We also compare the average MCE values observed in our ground measurements to MCE values reported for measurements from other platforms, including airborne measurements. Finally, we compare our average emission factors to values reported in the literature for other

ecosystems. Currently, highly cited compilations of emission factors (e.g., Akagi et al., 2011) do not include any results from Australian forests fires. In fact, the emission factors listed for temperate forests in Akagi et al. (2011) are sourced exclusively from measurements made at North American fires. We compare our results with the emission factors listed in Akagi et al. (2011, Table S4, February 2015 update) for temperate forests and to emission factors measured for Australian savanna fires and find significant differences in both cases.

## 2 Methods

### 2.1 Prescribed fires

Between 2010 and 2015, we sampled a total of nine prescribed fires in Australian temperate forests. Seven of those fires took place in New South Wales (NSW) in 2010-2013, the other two fires were sampled in the State of Victoria in April 2015. The locations of the fires sampled are indicated on the maps shown in Fig. 1. All fires took place in variants of dry sclerophyll

forests, dominated by eucalypt species. Table S1 lists the fires, their location, the dates on which they were sampled, the main vegetation type, the area burnt, the fuel loading, the time elapsed since the previous fire, the coordinates of the sampling sites and the method(s) of sampling deployed (these methods correspond to the colour coding on the maps in Fig. 1).

In NSW, all fires took place in the Greater Sydney area, as seen in Fig. 1. Dominant overstorey species included eucalypts (including *Eucalyptus*, *Corymbia* and *Angophora* species), with *Melaleuca*, *Acacia* and *Banksia* species in the sub-canopy and

the shrubby understorey. The ground cover was generally made up of native grasses and a litter of eucalypt leaves and twigs, as well as fallen tree limbs of varying sizes.

In Victoria, dominant canopy species were mostly eucalypts. Dominant overstorey species were *E. radiata* (Sieb. ex. DC.), *E. obliqua* (L'Hérit.), *E. dives* (Schau.), *E. leucoxylon* (F. Muell.) and *E. macrorhyncha* (F. Muell.). *Acacia* and *Banksia* species





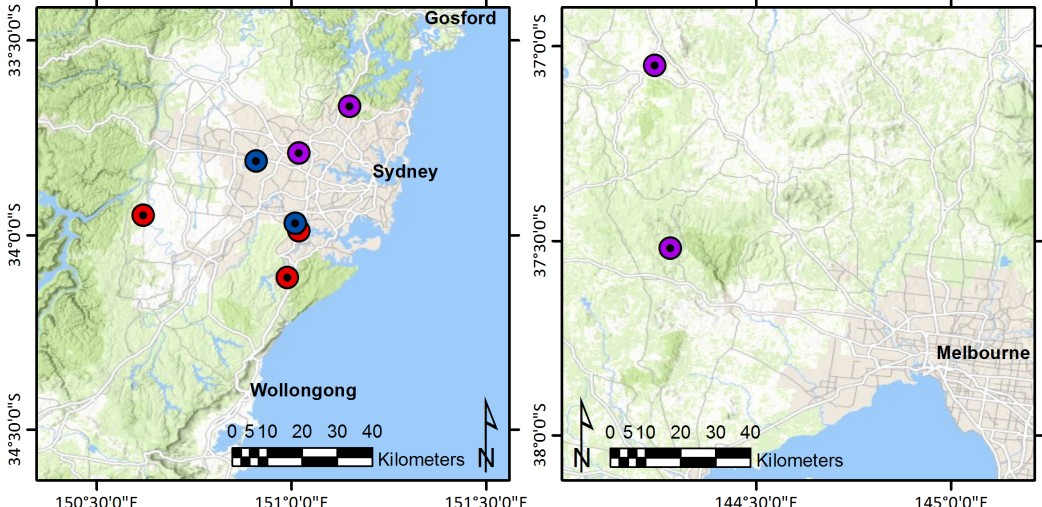

**Figure 1.** Locations of the nine prescribed fires in Australian temperate forests sampled between 2010 and 2015. The NSW fires are on the left, and the fires in Victoria on the right. The red dots represent fires where both open-path FTIR (OP-FTIR) and grab sampling took place, the blue dots indicate fires where only grab sampling took place, and the purple dots indicate fires where only OP-FTIR sampling took place.

dominated the understorey. Ground cover was dominated by tree litter, with gorse (*Ulex europaeus*) and blackberry (*Rubus fruticosus*) recorded in some areas.

## 2.2    Open-path FTIR system (OP-FTIR)

An open-path FTIR system was deployed at five prescribed fires in NSW and at the two prescribed fires in Victoria, as indicated
in the last column of Table S1. The system used in this project is described in detail in Paton-Walsh et al. (2014). Briefly, the spectrometer (Bomem MB-100 Series, 1 cm$^{-1}$ resolution) has a built-in infrared source and is placed 20-50 meters away from a set of retro-reflectors positioned so that smoke from the fire crosses the path in between. The system can run autonomously and records a spectrum consisting of three scans, approximately every twenty seconds. Typically, the system is set up and starts recording before the fire is ignited, and is left to run until mole fractions return to ambient values. As the measurement
is integrated over a path of several meters and is continuous over the duration of the fire, the emissions measured using this technique are likely to capture smoke from all stages of the fire, and therefore to be representative of the whole fire. One of the great advantages of OP-FTIR is that there is no sample capture, avoiding losses due to walls or sample lines.

In April 2015, the OP-FTIR was deployed at two prescribed burns in temperate forests in Victoria, several hundred kilometres away from the fires sampled in 2010-2013. The first fire, on April 13$^{th}$, was near Greendale, Victoria, and the second, on April
23$^{rd}$, was in Kalimna Park, Castlemaine, Victoria (see Fig. 1 for a map of the locations). At the Greendale fire, the spectrometer was positioned along a forest road and the retro-reflectors were installed 45 m away within the woodland area to be burned, so that both smoke and flames passed through the line of sight of the instrument. At the Castlemaine fire, both the spectrometer




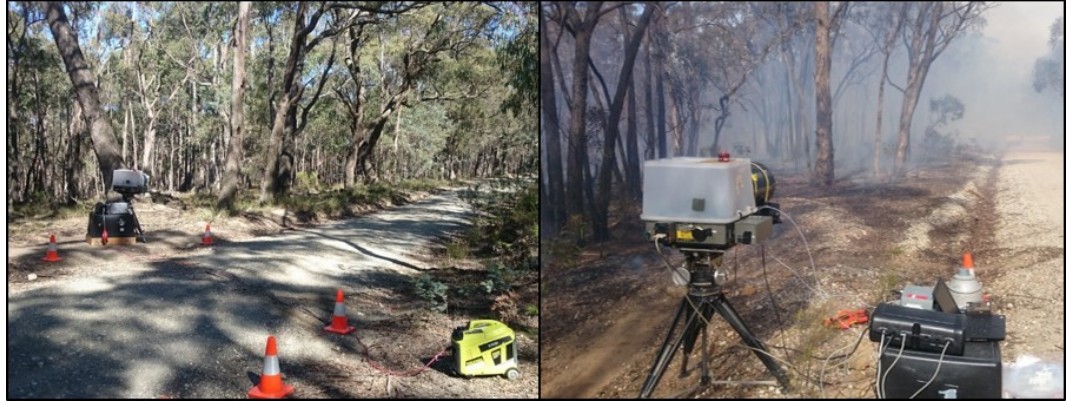

**Figure 2.** The instrumental set-up for the open-path FTIR measurements of smoke at Greendale on April $13^{th}$, 2015 (left) and Castlemaine on April $23^{rd}$, 2015 (right).

and the retro-reflectors were positioned along a forest track downwind of the fire, so that smoke would blow through the 50 m measurement path. The instrument set-up at both fires is shown in Fig. 2. The details of the NSW deployments are in Paton-Walsh et al. (2014).

The OP-FTIR spectra collected during the fires were subsequently analysed to derive mole fractions of $CO_2$, CO, $CH_4$, acetic acid, ammonia, ethene, formaldehyde, formic acid and methanol using the Multiple Atmospheric Layer Transmission (MALT) model (Griffith, 1996; Griffith et al., 2012) and the spectral windows described in Paton-Walsh et al. (2014).

## 2.3 Grab sampling

A total of 67 smoke samples were collected over seven days of sampling at five prescribed fires in NSW. Of those samples, over half were of well mixed, rising smoke. The others were from various targets, including smouldering litter and logs and burning grass and shrubs. The number of samples collected at each fire is indicated in brackets in the last column of Table S1. Samples were collected in 600 ml glass flasks, except at the Gulguer Plateau fire, where samples were collected into 1 L Tedlar bags. The glass flasks were pre-evacuated using a turbo-molecular pump (Pfeiffer TCS 010) prior to deployment to the fires, and filled with smoke on site by opening them for a few seconds. The bags were flushed with high purity nitrogen and brought to the Gulguer Plateau fire where they were filled with smoke using a differential pressure system or 'vacuum box' powered by a generator. Filling the bags took a few minutes, and consequently, most samples were collected from large smouldering targets after the fire front had moved through the sampling area.

All grab samples were brought back to the lab and analysed within 24 hours of collection. A Fourier Transform Infrared (FTIR) spectrometer coupled to a White cell was used to measure $CO_2$, CO, $CH_4$, ethane and ethene. VOC mole fractions were measured using selective ion flow tube mass spectrometry (SIFT-MS).



### 2.3.1 Fourier Transform Infrared (FTIR) spectrometer coupled to a White cell (White cell FTIR)

Mole fractions of $CO_2$, CO, $CH_4$, ethane and ethene in the grab samples of smoke collected at the fires were measured using a Bomem MB-100 Series FTIR spectrometer (1 cm$^{-1}$ resolution). This spectrometer is coupled to a multi-pass optical (White) cell with a path of 22.2 m and is fitted with a InSb detector cooled with liquid nitrogen.

Part of the sample was transferred to the evacuated White cell and the temperature and pressure inside the cell were logged. Typical temperatures and pressures inside the White cell were 22°C and 220 hPa, respectively. A spectrum was acquired for each grab sample by co-adding 78 scans. As for the OP-FTIR spectra, mole fractions were retrieved using the Multiple Atmospheric Layer Transmission (MALT) model (Griffith, 1996; Griffith et al., 2012).

### 2.3.2 Selective Ion Flow Tube Mass Spectrometry (SIFT-MS)

SIFT-MS is a technique for the on-line analysis of gas samples that is akin to the better-known Proton-Transfer-Reaction Mass Spectrometry (PTR-MS) (Blake et al., 2009). Both instruments use chemical ionization to ionize the VOCs present in air and both are equipped with quadrupole mass filters. The main advantage of SIFT-MS is its capability to switch between three reagent ions ($H_3O^+$, $NO^+$ and $O_2^+$) within a single measurement cycle, allowing the detection of species such as acetylene and ethene in addition to the species commonly detected using PTR-MS within the same analysis. It does this by producing all three

reagent ions simultaneously in a microwave discharge and then selecting one or the other (switching) using a quadrupole mass filter (the instrument therefore has two quadrupole mass filters). By contrast, PTR-MS is typically equipped with a hollow-cathode discharge that produces a pure stream of a single reagent ion (most commonly $H_3O^+$) and therefore requires a single quadrupole. Another difference is that PTR-MS uses a drift tube as its reaction chamber (in which ions are carried by an electric field), whereas SIFT-MS is equipped with a flow tube. The specific instrument used in this study (Syft Voice 100) uses a stream

of helium and argon to thermalize and carry the ions (Milligan et al., 2007). This means that the instrument dilutes the sample by a factor that is a function of the pressure and temperature inside the flow tube, and of the flows of sample and carrier gases. This makes the instrument less sensitive than PTR-MS (Blake et al., 2009) but ideally suited for the analysis of highly polluted air, such as smoke samples. The flow tube dilution ratio in this study was about 1:15.

The SIFT-MS was operated in multiple ion mode, targeting eighteen VOC species. Table S2 lists the species targeted,

the reagent ion used, the mass-to-charge ratios measured and the calibration factors used to quantify them. The list includes aromatic species, nitrogen-containing species, some oxygenated species, some small hydrocarbons and some biogenic species, targeting a breadth of chemical classes. The species targeted were for the most part the most abundant reported at their nominal molecular mass by Yokelson et al. (2013), who deployed extensive instrumentation in a laboratory setting and calculated emission factors for 357 species. A notable exception is the signal at $NO^+$ 68, which is calibrated using isoprene, but is

expected to be dominated by furan in smoke samples. Also, the signal at $H_3O^+$ 71 is expected to include 2-butenal as well as methacrolein and methyl vinyl ketone. The measurement cycle took approximately 7 seconds to complete and was repeated 8 times on each smoke sample. Mole fractions of VOCs were computed from raw SIFT-MS spectra using the calibration factors listed in Table S2. For each sample, an average mole fraction was calculated for each species by taking the mean over all





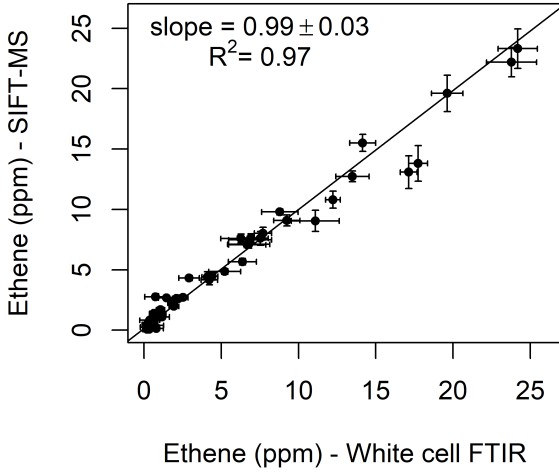

**Figure 3.** Comparison of ethene mole fractions measured by SIFT-MS with those measured by White cell FTIR in grab samples of smoke collected at Australian temperate forest fires.

repeats. An average mole fraction was reported for a given species only if its signal-to-noise ratio was greater than three, i.e. if the average signal was at least three times greater than the standard deviation of its mean.

The linearity of the SIFT-MS response was checked by plotting the mole fractions measured for ethene against those measured by White cell FTIR in the same grab samples. Figure 3 shows the good agreement for ethene between the two methods. The plot demonstrates that there was no loss of linearity in the SIFT-MS response even at high mole fractions, which is a result of the sample dilution that occurs within the flow tube of the instrument.

### 2.4 Determination of emission ratios (ER)

Emission ratios (ER) were derived by plotting VOC mole fractions against those of CO or $CO_2$ (or another reference VOC species in some cases, see below) and applying an orthogonal regression. Orthogonal regression finds the best line of fit by minimising squared distances between (x, y) points and their projection on the line of best fit. The regression is also weighted by the uncertainties in both x and y, which, in this case, are the measurement uncertainties.

For the grab samples, emission ratios were derived for individual fires when possible; however, the VOC results from the targeted grab sampling were more highly variable than the open-path measurements in the well-mixed smoke, as is common for this type of sampling (Yokelson et al., 2008, 2013; Burling et al., 2011; Akagi et al., 2013). This resulted in poor correlations ($R^2 < 0.5$) for some species for certain fires. Also, not every trace gas species was present at a detectable level in every sample. For some fires, this resulted in too few samples to allow an emission ratio to be meaningfully derived by regression for that





species. For this reason, emission ratios for each species were also derived through combining samples from all fires. Certain VOC species did not correlate strongly with either CO or $CO_2$. In those cases, emission ratios were derived using another reference species, e.g. an emission ratio to acetonitrile was derived for pyrrole, and ethene was used as a reference species to derive an emission ratio for benzene, 1,3-butadiene and acetylene.

## 2.5  Determination of emission factors (EF)

An emission factor (EF) is defined as the mass of trace gas of interest (X) released per amount of dry biomass burnt and is typically expressed in units of g kg$^{-1}$:

$$EF_X = 1000 \times \frac{mass_X}{mass_{dryfuelburnt}} \quad (1)$$

This is a very direct method of estimating emissions, but can only be used if all the emissions are captured (so that the total
mass of gas X can be measured) and if the mass of biomass burnt in the fire is known (Andreae and Merlet, 2001), which is rarely the case except in laboratory experiments. In the absence of such knowledge, the total mass of biomass burnt can be derived from the total mass of carbon emitted and the fractional carbon content of the biomass burnt ($F_{carbon}$), which is sometimes measured but often estimated:

$$EF_X = F_{carbon} \times 1000 \times \frac{mass_X}{mass_{dryfuelburnt}} \quad (2)$$

In this study, $F_{carbon}$ was assigned a value of 0.5, as in Akagi et al. (2011),Yokelson et al. (2011) and Paton-Walsh et al. (2014). Similarly, the total mass of carbon emitted by a fire is usually not known, and is estimated by measuring the most abundant carbon-containing species emitted by the fire. The emission factor for species X is then:

$$EF_X = F_{carbon} \times 1000 \times \frac{MM_X}{12} \times \frac{C_X}{C_T} \quad (3)$$

where $MM_X$ is the molecular mass of the species of interest, 12 is the atomic mass of carbon and $\frac{C_X}{C_T}$ is the number of moles
of species X emitted divided by the total number of moles of carbon emitted. In general, only a subset of the smoke from a fire is sampled. If that sample is representative of the whole fire, then the observed ratio of a species to the sum of all other species $\frac{C_X}{C_T}$ should be representative of the entire fire. $\frac{C_X}{C_T}$ can be calculated directly from the excess amounts measured:

$$EF_X = F_{carbon} \times 1000 \times \frac{MM_X}{12} \times \frac{\Delta[X]}{\sum_{y=1}^{n} NC_y \times \Delta[Y]} \quad (4)$$





where $\Delta[X]$ and $\Delta[Y]$ are the total excess mole fraction of the species of interest and of another carbon-containing species, respectively, $NC_y$ is the number of carbon atoms in species Y and the sum is over all carbon-containing species measured in the smoke. Equation 4 can also be written as:

$$EF_X = F_{carbon} \times 1000 \times \frac{MM_X}{12} \times \frac{ER_{X/ref}}{\sum_{y=1}^{n} NC_y \times ER_{Y/ref}} \tag{5}$$

and it follows that the emission factor for a given species of interest can be calculated from the emission ratio of that species to the reference species, and the emission factor of the reference species:

$$EF_X = ER_{X/ref} \times \frac{MM_X}{MM_{ref}} \times EF_{ref} \tag{6}$$

There are variants on how to apply the equations above, see Paton-Walsh et al. (2014) for a discussion. In this project, we chose the same approach as in Paton-Walsh et al. (2014) to process the open-path FTIR data and calculated emission factors for CO and $CO_2$ using Eq. 4 with $\frac{C_X}{C_T}$ calculated using the total excess amounts of each gas detected by summing over the excess amounts from each measurement. The emission factors of other species were calculated using Eq. 6.

Emission factors for CO, $CO_2$ and $CH_4$ were calculated for each individual grab sample using Eq. 4, with $C_T$ calculated as the sum of $CO_2$, CO and $CH_4$ only. These were then used with Eq. 6 to derive emission factors for individual fires. To determine study-average emission factors from the grab sample data, we used Eq. 6 and the emission factors for CO and $CO_2$ derived from the in situ OP-FTIR measurements at the NSW fires. If the emission ratio for a given VOC was derived using another VOC (instead of CO or $CO_2$), their emission ratio was first converted to an emission ratio to CO or $CO_2$ using the emission ratio of their reference VOC to CO or $CO_2$. The uncertainty on the resulting emission ratio to CO (or $CO_2$) was calculated by adding the uncertainties in quadrature.

## 2.6 Calculation of modified combustion efficiency (MCE)

MCE is a proxy for combustion efficiency, which is defined as the proportion of total carbon emitted by a fire released as $CO_2$. MCE is defined as the excess mole fraction of $CO_2$ divided by the sum of the excess mole fractions of $CO_2$ and CO (Hao and Ward, 1993; Yokelson et al., 1996):

$$MCE = \frac{\Delta CO_2}{\Delta CO_2 + \Delta CO} \tag{7}$$

When the fire is dominated by flaming combustion, the modified combustion efficiency is high, meaning that the emissions are dominated by $CO_2$. The combustion efficiency decreases as smouldering combustion and emissions of CO become more dominant. Flaming combustion is generally associated with MCE values greater than 0.9 and smouldering combustion with values below 0.9 (Yokelson et al., 1996; Bertschi et al., 2003).





In this project, the MCE of a fire sampled by OP-FTIR was determined from the total excess amounts of $CO_2$ and CO detected by the open-path system (i.e. by summing the excess amounts from each measurement recorded). For grab samples, MCE was calculated for each individual sample using Eq. 7. These are indicative of the type of combustion (e.g. flaming vs. smouldering) captured by the grab sampling, and are not necessarily representative of the whole fire. As an example, the
average MCE of the grab samples collected at the Gulguer Plateau fires was $0.78 \pm 0.09$ whereas a fire-integrated value of 0.90 was measured by OP-FTIR (Paton-Walsh et al., 2014).

## 3  Results

### 3.1  Emission ratios and emission factors determined from grab samples collected at prescribed fires in NSW and analysed using SIFT-MS and White-cell FTIR

Emission ratios (ER) were derived for all species measured in the grab samples by White cell FTIR and SIFT-MS as per Sect. 2.4. Emission ratios for individual fires, when available, are listed in Table S3. Table 1 lists the emission ratios derived from combining data from all fires ('all data combined'). When emission ratios for individual fires are available (see Table S3), the mean emission ratio is also included in Table 1. Figure S1 shows the correlation of ethane with CO for each of the five individual fires, and for all fires combined, as an example. Figure 4 shows the correlations for six species (hydrogen cyanide,
formaldehyde, acetylene, pyrrole, monoterpenes, and the sum of $C_8H_{10}$ species) for which only an 'all data combined' emission ratio could be derived.

The emission ratios of some species show important site-to-site variability (see Table S3). For example, the emission ratio of $CH_4$ to CO measured at Prospect Reservoir is lower than the average (0.06 (0.01), see Table S3). The site at Prospect Reservoir was mostly grassy, and the emission ratio measured there ($0.037 \pm 0.004$) is close to the one measured in tussock-
and hummock-grass savanna open woodland fires in northern Australia ($0.040 \pm 0.007$) by Smith et al. (2014).

Similarly, the emission ratio of acetonitrile to CO is markedly lower at Gulguer Plateau than at the other fires. This could be due to the lower nitrogen content of logs compared to foliage and twigs (Susott et al., 1996; Snowdon et al., 2005), resulting in lower emissions of nitrogen-containing species (Coggon et al., 2016). The emission ratio measured for acetonitrile at Gulguer Plateau is excluded from the mean emission ratio listed in Table 1. Including this emission ratio reduces the mean ER from 0.05
$\pm 0.01$ to $0.04 \pm 0.02$. The Gulguer fire is also excluded from the emission ratio for acetonitrile derived from combining data from all fires, since including it results in $R^2 < 0.5$. Figure 5 shows the correlations of acetonitrile with CO; the Gulguer Plateau fire is shown in red, the other four fires are shown in black. Pyrrole showed the same behaviour against CO as acetonitrile. Its emission ratio was therefore derived to acetonitrile instead of CO.

Despite this site-to-site variability in the emission ratio of certain species, the mean emission ratio is usually the same,
within the uncertainties, as the value derived from combining samples from all fires. This indicates that the 'all data combined' emission ratios listed in Table 1 should be similarly representative of the ecosystem sampled - a useful result since this is the only ER available for some species. Whole-fire emission factors were then calculated using the 'all data combined' emission ratios listed in Table 1 and the average fire-integrated emission factors for CO and $CO_2$ measured by OP-FTIR at the NSW





**Table 1.** Summary of emission ratios (ER) determined for species measured by SIFT-MS and White cell FTIR in grab samples collected at the NSW fires. Mean ER is the average ER measured at individual fires. The "all data combined" ER was derived using a linear regression.

| Species | Reference species | Mean ER (std. dev.) | ER (all data combined) | # of samples | $R^2$ |
|---|---|---|---|---|---|
| **White cell FTIR** | | | | | |
| CO | $CO_2$ | 0.19 (0.15) | $0.17 \pm 0.06$ | 67 | 0.47 |
| $CH_4$ | CO | 0.06 (0.01) | $0.059 \pm 0.003$ | 67 | 0.89 |
| Ethane | CO | 0.004 (0.001) | $0.0038 \pm 0.0003$ | 67 | 0.87 |
| Ethene | $CO_2$ | | $0.0017 \pm 0.0002$ | 58 | 0.71 |
| **SIFT-MS** | | | | | |
| Ethene | $CO_2$ | | $0.0018 \pm 0.0002$ | 54 | 0.77 |
| Acetaldehyde | CO | 0.009 (0.002) | $0.007 \pm 0.001$ | 50 | 0.75 |
| Acetone | CO | 0.005 (0.002) | $0.0034 \pm 0.0005$ | 47 | 0.74 |
| Acetonitrile[a] | CO | 0.0039 (0.0008) | $0.0038 \pm 0.0005$ | 42 | 0.91 |
| Acetylene | Ethene | | $0.21 \pm 0.04$ | 29 | 0.59 |
| Benzene | Ethene | 0.08 (0.01) | $0.078 \pm 0.006$ | 43 | 0.84 |
| Butadiene | Ethene | 0.042 (0.006) | $0.042 \pm 0.002$ | 38 | 0.95 |
| Butanone | CO | | $0.00082 \pm 0.00007$ | 45 | 0.69 |
| Ethanol[b] | CO | | $0.00021 \pm 0.00005$ | 7 | 0.97 |
| Formaldehyde | Hydrogen cyanide | | $2.9 \pm 0.3$ | 50 | 0.65 |
| Furan + isoprene | CO | 0.0018 (0.0006) | $0.0019 \pm 0.0003$ | 37 | 0.87 |
| Hydrogen cyanide | CO | | $0.0063 \pm 0.0007$ | 50 | 0.46 |
| sum of MACR, MVK and 2-butenal | CO | | $0.0035 \pm 0.0009$ | 44 | 0.73 |
| Methanol | CO | 0.025 (0.006) | $0.022 \pm 0.002$ | 54 | 0.72 |
| Monoterpenes | Methanol | | $0.042 \pm 0.006$ | 33 | 0.86 |
| Pyrrole | Acetonitrile | | $0.15 \pm 0.07$ | 25 | 0.78 |
| Toluene | CO | 0.0006 (0.0002) | $0.0006 \pm 0.0001$ | 40 | 0.75 |
| sum of $C_8H_{10}$ species | Toluene | | $0.42 \pm 0.04$ | 36 | 0.75 |

[a] values reported exclude the Gulguer Plateau fire - see text for detail

[b] value reported is for the Alfords Point fire




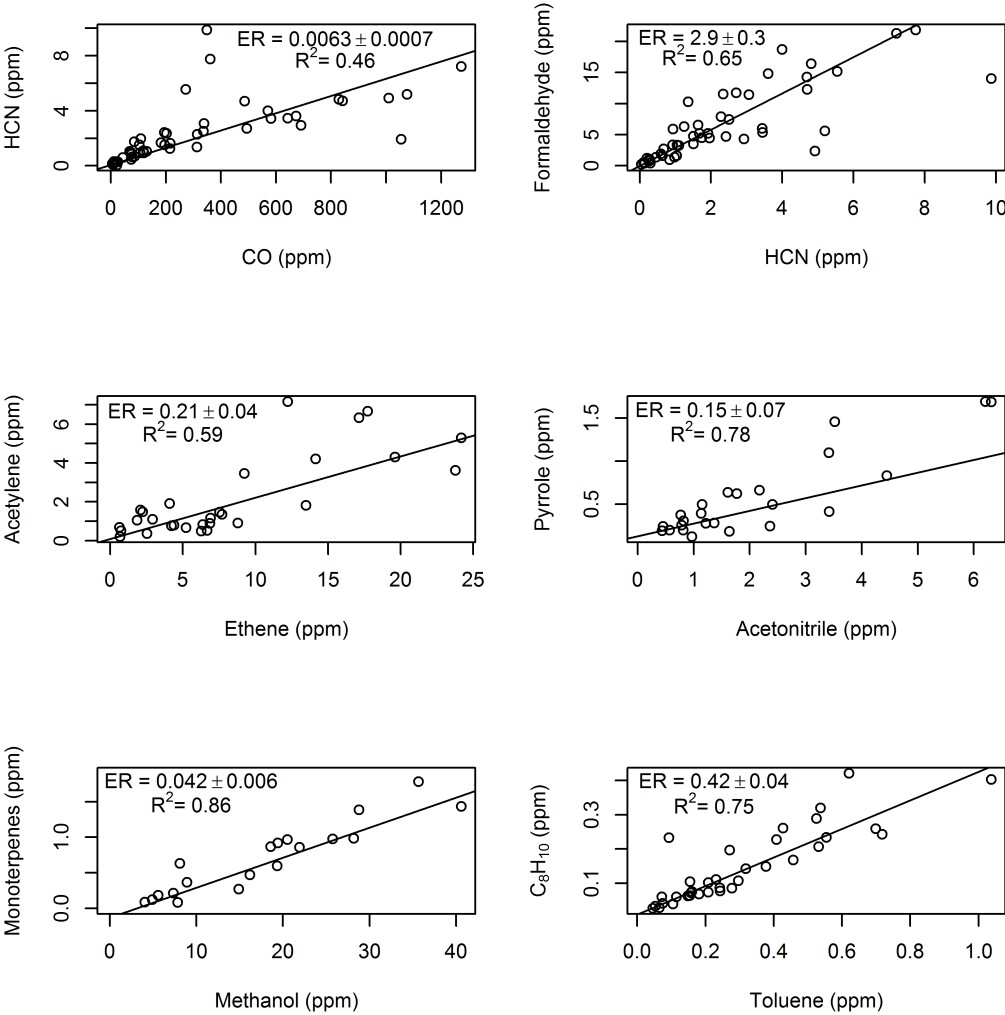

**Figure 4.** Examples of correlations for species for which only a single 'all data combined' emission ratio (ER) could be derived from the grab sample measurements. Top left is hydrogen cyanide (HCN) to CO, top right is formaldehyde to HCN, middle left is acetylene to ethene, middle right is pyrrole to acetonitrile, bottom left is monoterpenes to methanol and bottom right is the sum of $C_8H_{10}$ species to toluene.





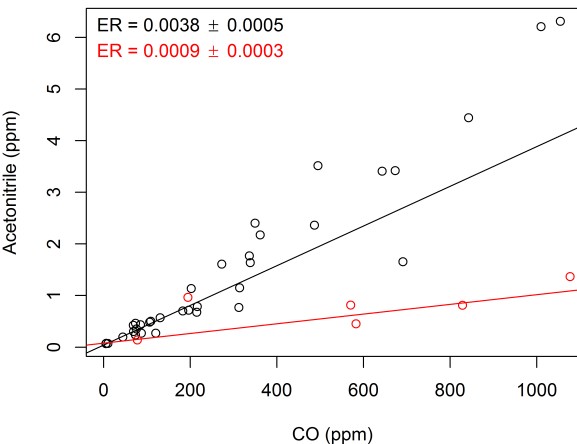

**Figure 5.** Emission ratio (ER) for acetonitrile to CO for the Gulguer Plateau fire grab samples (in red) and for the other four fires (in black).

fires by Paton-Walsh et al. (2014) and reproduced in the last column of Table 2. The resulting whole-fire emission factors for all VOC species are listed in Table 5.

## 3.2 Open-path FTIR results from prescribed fires in temperate forests in Victoria

All trace gases measured by OP-FTIR at the prescribed fires in Victoria exhibited strong correlations with either CO or $CO_2$.
Correlations between the measured species at the Castlemaine fire are shown in Figure S2 as an example. The calculated emission ratios and emission factors are listed in Table 2. Uncertainties were calculated as per Appendix B of Paton-Walsh et al. (2014).

There is little variability seen between the two fires sampled in Victoria. The emission ratios measured at the two fires are comparable, and the emission factors agree within their uncertainties. The emission ratios measured in Victoria are within the
10 range of values measured at the NSW fires for all species except formic acid and acetic acid (Table 2). The average observed MCE of 0.92 at the Victorian fires is higher than that reported by Paton-Walsh et al. (2014) for the NSW fires (average 0.90, range: 0.88-0.91). The emission factors listed in Table 2 generally reflect this difference, with species typically associated with smouldering combustion having slightly lower emission factors at the Victorian fires. The differences are slight however, and the emission factors measured at the fires in Victoria generally agree within the stated uncertainties with those reported for
the NSW fires. One major exception is acetic acid. Its emission ratio at the fires in Victoria was double that seen at the NSW fires, and this is reflected in the emission factors. This indicates a difference in emissions from the different regions sampled that is not explained by the difference in modified combustion efficiency. The dependence of emission factors derived from the OP-FTIR measurements on MCE is explored more fully in the next section.



**Table 2.** Summary of open-path FTIR measurements at prescribed fires in temperate forest in the State of Victoria and comparison with similar results obtained at prescribed fires in New South Wales. Values in parentheses are standard deviations of the mean.

| Species | Reference species | Castlemaine ER | $R^2$ | EF | Greendale ER | $R^2$ | EF | NSW fires[a] ER | EF |
|---|---|---|---|---|---|---|---|---|---|
| $CO_2$ | | | | $1650 \pm 170$ | | | $1670 \pm 170$ | | $1620\ (160)$ |
| CO | | | | $101 \pm 16$ | | | $84 \pm 13$ | | $118\ (19)$ |
| $CH_4$ | CO | $0.0571 \pm 0.0006$ | 0.97 | $3.3 \pm 0.2$ | $0.0633 \pm 0.0005$ | 0.99 | $3.1 \pm 0.2$ | $0.05\ (0.01)$ | $3.6\ (1.1)$ |
| Ammonia | CO | $0.0276 \pm 0.0003$ | 0.98 | $1.7 \pm 0.2$ | $0.0291 \pm 0.0004$ | 0.95 | $1.5 \pm 0.2$ | $0.021\ (0.008)$ | $1.6\ (0.6)$ |
| Ethene | $CO_2$ | $0.00118 \pm 0.00001$ | 0.97 | $1.2 \pm 0.3$ | $0.00105 \pm 0.00002$ | 0.91 | $1.1 \pm 0.2$ | $0\ .0012\ (0.0003)$ | $1.3\ (0.3)$ |
| Formaldehyde | $CO_2$ | $0.00133 \pm 0.00002$ | 0.94 | $1.5 \pm 0.3$ | $0.00113 \pm 0.00003$ | 0.82 | $1.3 \pm 0.2$ | $0.0016\ (0.0004)$ | |
| Methanol | CO | $0.0144 \pm 0.0002$ | 0.96 | $1.7 \pm 0.3$ | $0.0154 \pm 0.0006$ | 0.95 | $1.5 \pm 0.4$ | $0.017\ (0.006)$ | $2.4\ (1.2)$ |
| Formic acid | CO | $0.00321 \pm 0.00005$ | 0.94 | $0.5 \pm 0.2$ | $0.00414 \pm 0.00007$ | 0.93 | $0.6 \pm 0.1$ | $0.0021\ (0.0007)$ | $0.4\ (0.2)$ |
| Acetic acid | CO | $0.0303 \pm 0.0003$ | 0.98 | $6.5 \pm 1.2$ | $0.0331 \pm 0.0005$ | 0.95 | $6.0 \pm 0.9$ | $0.015\ (0.003)$ | $3.8\ (1.3)$ |

[a] Paton-Walsh et al. (2014)

## 3.3 Dependence of emission factors of trace gases from Australian temperate forest fires on modified combustion efficiency (MCE)

The MCE dependence of the emissions of carbon-containing species from all fires sampled using OP-FTIR as part of this ground-based study is explored in this section. The emission factors calculated for each fire sampled by OP-FTIR are plotted

5   as a function of fire-averaged MCE in Fig. 6. The regression statistics are listed in Table 3. As the range of observed MCE is relatively narrow, the relationship is well represented using a linear regression. For larger MCE ranges, an exponential fit may be more appropriate (e.g. Meyer et al. (2012) suggest an exponential fit for $CH_4$).

The magnitude of the slope and the intercept listed in Table 3 reflects the magnitude of the emission factor for that species. The strength of the relationship is judged from the coefficient of determination ($R^2$) and the p-value.

10   For some species, there is no significant relationship with MCE when including data from all seven fires. This is the case for formic acid and acetic acid, for which significantly different emission ratios were measured at the fires in Victoria. Similarly,





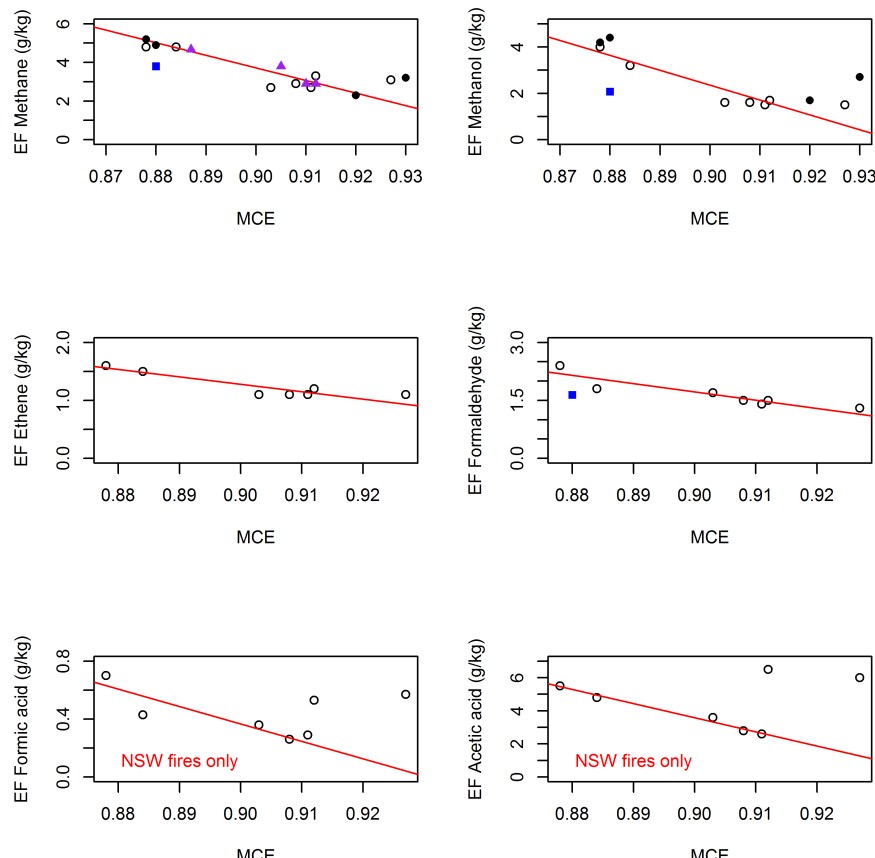

**Figure 6.** Dependence of emission factors on MCE. Open circles represent the seven fires sampled using OP-FTIR with the line of best fit shown in red. For formic acid and acetic acid, this regression line was derived using the measurements from the NSW fires only. The black circles represent average results from grab samples at four fires (the Gulguer Plateau fire falls outside the range of MCE values measured by OP-FTIR and is not shown). Purple triangles represent the methane results from the airborne measurements of Hurst et al. (1996) and the blue squares represent the emission factors measured for methane, methanol, and formaldehyde by Lawson et al. (2015) in a transported plume impacting the Cape Grim Baseline Air Pollution Station in Tasmania.





**Table 3.** Summary of emission factor dependence on modified combustion efficiency (MCE)

| Species | Data used in regression calculation | Slope | Intercept | $R^2$ | p value |
|---|---|---|---|---|---|
| $CH_4$ | NSW and VIC fires | $-65 \pm 20$ | $62 \pm 17$ | 0.61 | 0.02 |
| Ethene | NSW and VIC fires | $-13 \pm 4$ | $13 \pm 3$ | 0.75 | 0.007 |
| Formadehyde | NSW and VIC fires | $-21 \pm 10$ | $21 \pm 9$ | 0.79 | 0.005 |
| Methanol | NSW and VIC fires | $-64 \pm 16$ | $60 \pm 14$ | 0.79 | 0.005 |
| Formic acid | NSW fires only | $-12 \pm 6$ | $11 \pm 5$ | 0.74 | 0.04 |
| Acetic acid | NSW fires only | $-86 \pm 5$ | $81 \pm 4$ | 0.98 | 0.004 |
| sum of furan and isoprene | Grab samples | $-9 \pm 5$ | $9 \pm 4$ | 0.95 | 0.005 |
| sum of acetone and propanal | Grab samples | $-5 \pm 2$ | $6 \pm 1$ | 0.94 | 0.009 |

the emission factor for $CH_4$ has a stronger relationship with MCE when considering only the NSW fires. This indicates that combustion efficiency is not the only factor that controls differences in emissions for these species.

The emission factors measured by Hurst et al. (1996) for $CH_4$ and Lawson et al. (2015) for $CH_4$, methanol and formaldehyde are also included in Fig. 6 for comparison. Figure 6 also includes the average results derived from the grab samples for both

$CH_4$ and methanol for each fire except the Gulguer Plateau fire. The reasonable agreement with the OP-FTIR measurements seen in Fig. 6 means that it may be possible to estimate the MCE-dependence of the species that were only measured in grab samples (by SIFT-MS or White-cell FTIR). For this analysis, average values from the five fires were used, spanning a range of average MCE of 0.78 to 0.93. No statistically significant trend was found for acetaldehyde, acetonitrile, benzene, butadiene, ethane and toluene, but there were significant trends for the sum of furan and isoprene and for the sum of acetone and propanal.

The statistics for these trends are listed in Table 3. The MCE dependence of the other measured species could not be determined because fire-specific emission ratios were not available.

## 4   Discussion

### 4.1   Comparison with MCE-dependent emission factors from North American temperate forests

The MCE dependence of emission factors listed in Table 3 were compared to those reported by Akagi et al. (2013) for fires in

conifer forests in South Carolina, and by Burling et al. (2011) for fires in conifer forests in North Carolina and for chaparral fires in California. There is considerable variability between the two North American studies, even for the similar conifer ecosystems sampled. Both studies found negative relationships to MCE for $CH_4$ (with slopes ranging from $-65 \pm 13$ to $-96 \pm 10$), methanol (with slopes ranging from $-21 \pm 6$ to $-39 \pm 2$) and furan ($-6 \pm 3$ to $-8 \pm 1$). These results are consistent with the





ones listed in Table 3 for these species, although the slope measured in Australian temperate forests for methanol is larger (-64 ± 16).

For other species, the results are mixed, with Akagi et al. (2013) finding no relationship to MCE for acetic acid but Burling et al. (2011) finding a strong one (with a slope of -45 ± 3 and $R^2$ = 0.98) in a similar conifer ecosystem. This is analogous to the results presented here, where a strong relationship to MCE is found for a subset of the data (NSW fires only, slope = -86 ± 5, $R^2$ = 0.98) but no relationship is found when all the fires are considered. For formic acid, both North American studies find a relationship for conifer forest fires (with slopes of -1.8 ± 0.6 and -3.1 ± 0.2), but Burling et al. (2011) found no relationship for chaparral fires. In this study, we find a relationship for the NSW fires, but no relationship when including all fires.

For formaldehyde and ethene, Akagi et al. (2013) reports a weak or insignificant relationship to MCE whereas Burling et al. (2011) reports strong relationships to MCE for both species for fires in a similar conifer ecosystem (with slopes of -21 ± 2 for formaldehyde and -11 ± 2 for ethene) and a weak or insignificant relationship to MCE for fires in chaparral. For fires in Australian temperate forests, we observed similar slopes of -21 ± 10 for formaldehyde and -13 ± 4 for ethene.

Akagi et al. (2013) report a slope of -16 ± 4 for acetone, which is larger than the one observed for the sum of acetone and propanal in this study (-5 ± 2). Akagi et al. (2013) also report significant relationships to MCE for ethane, benzene, toluene, xylenes, acetonitrile and acetaldehyde whereas no relationship was observed for these species in our study.

Considering the variability of relationships to MCE observed even for similar ecosystems, it seems likely that other factors are influencing emissions, especially of those species that are biogenically produced by vegetation and are not only a product of combustion. This limits the usefulness of MCE as a means of extrapolating emission factors for these species. Nevertheless, the MCE measured at a fire can be a good indication of whether a representative sample has been captured. This is explored in the next section by comparing MCE values observed from different measurement platforms for Australian temperate forest fires.

## 4.2 Comparison of MCE, $CO_2$, CO and $CH_4$ emission factors measured for Australian temperate ecosystems from various platforms

MCE and emission factors for $CO_2$, CO and $CH_4$ for Australian temperate ecosystems have been measured from a variety of platforms, including airborne measurements (Hurst et al., 1996) and measurements of plumes transported short distances to fixed monitoring stations (Lawson et al., 2015; Rea et al., 2016). Comparing these results to our ground-based measurements (see Table 4) reveals that there is a relatively small spread of MCE values measured for fires in Australian temperate ecosystems. Even airborne measurements over the very large Sydney wildfires of January 1994 have a relatively low average MCE of 0.91 (Hurst et al., 1996). Furthermore, there is no significant difference in the MCE observed for wild or prescribed fires, or between measurement platforms (Kruskal-Wallis rank sum test, p > 0.7). This is in contrast with measurements conducted at prescribed fires in North America, where higher average MCE values were observed for airborne measurements than for open-path measurements on the ground (0.93 vs. 0.91 on average for the same fires in Akagi et al. (2014), for example). MCE values of 0.93 or greater for airborne measurements have also been reported by other US studies (Burling et al., 2011; Akagi et al., 2013). The top left panel of Fig. 6 shows the $CH_4$ emission factors reported by Hurst et al. (1996) plotted alongside the



**Table 4.** Comparison of whole-fire modified combustion efficiency (MCE) and whole-fire emission factors for $CO_2$, CO and $CH_4$ reported in the literature for fires in Australian temperate forests and temperate forests in North America.

| Study | Location | MCE | EF $CO_2$ | EF CO | EF $CH_4$ | Platform | Type of fire |
|---|---|---|---|---|---|---|---|
| Hurst et al. (1996)[a] | Helensburgh, NSW, Australia | 0.91 | 1577 | 99 | 2.9 | Airborne | Wildfire |
| | Worragee, NSW, Australia | 0.89 | 1540 | 125 | 4.7 | Airborne | Wildfire |
| | Sydney, NSW, Australia | 0.91 | 1558 | 104 | 3.8 | Airborne | Wildfire |
| | Bateman's Bay, NSW, Australia | 0.91 | 1577 | 97 | 2.9 | Airborne | Prescribed fire |
| Lawson et al. (2015) | Robbin Island, TAS, Australia | 0.88 | 1621 | 127 | 3.8 | Transported plume | Wildfire |
| Paton-Walsh et al. (2014) | Greater Sydney Area, NSW, Australia | 0.90 (0.2) | 1620 (160) | 118 (19) | 36 (1.1) | Ground-based OP-FTIR | Prescribed fires |
| Rea et al. (2016) | Greater Sydney Area, NSW, Australia | 0.91 | 1640 | 107 | 7.8[b] | Transported plume | Wildfires |
| This study | Central Highlands, VIC, Australia | 0.92 (0.01) | 1660 (170) | 93 (15) | 3.2 (0.2) | Ground-based OP-FTIR | Prescribed fires |
| Akagi et al. (2011)[c] | North America | – | 1647 (37) | 88 (19) | 3.4 (0.9) | Mixed | Smoke <20 min |

[a] Hurst et al. (1996) assume 6 % of carbon is emitted as ash, which explains the lower emission factors reported for $CO_2$

[b] this value may be influenced by other sources - see Rea et al. (2016)

[c] Table S4, February 2015 update

OP-FTIR measurements conducted as part of this study and as part of Paton-Walsh et al. (2014). The agreement between the two platforms is excellent.

The average emission factor measured for $CH_4$ in Australian temperate forests is 3.5 (0.8) g kg$^{-1}$ dry fuel burnt (this value excludes the emission factor reported by Rea et al. (2016) as it may have been influenced by other sources). The average for the ground-based OP-FTIR measurements is 3.5 (0.9) g kg$^{-1}$ dry fuel burnt. These are in excellent agreement with the emission factor for $CH_4$ of 3.4 (0.9) g kg$^{-1}$ dry fuel burnt listed for temperate forests in Akagi et al. (2011, Table S4, February 2015 update).




### 4.3 Comparison of VOC emission ratios and emission factors measured for temperate ecosystems

Measurements of VOC emission factors have been more limited for Australian temperate forests. Enhancement ratios to CO for methanol, ammonia, formic acid, formaldehyde, acetylene, ethene and ethane were measured in lofted plumes from wildfires

by ground-based solar remote sensing Fourier transform spectrometry (Paton-Walsh et al., 2005, 2008) and satellite-based spectroscopic measurements (Young and Paton-Walsh, 2011; Glatthor et al., 2013). These were compared to the emission ratios measured in fresh smoke by OP-FTIR in NSW in Paton-Walsh et al. (2014) and this discussion is not repeated here.

The only other study to have reported emission factors for a significant number of trace gas species is that of Lawson et al. (2015). They report emission ratios and emission factors for trace gases and aerosol from opportunistic measurement of a

biomass burning plume impacting Cape Grim Baseline Air Pollution Station in Tasmania in February 2006. The plume was advected to the Station from a fire on a nearby island, mostly at night (from 23:00 AEST until 09:00 AEST). Their emission ratios and emission factors for VOCs are listed alongside ours in Table 5. Emission factors from Akagi et al. (2011, Table S4, February 2015 update) are also included for comparison. For some of the species measured by SIFT-MS in this study and by PTR-MS in Lawson et al. (2015), the reported emission factors are sum measurements of several species, including

potential contributions from unidentified compounds. In these cases, the emission factors of all species that could contribute were sourced from Akagi et al. (2011, Table S4, February 2015 update) and listed in the last column of Table 5.

There is considerable variability in the emission factors listed in 5, and most species agree within their stated uncertainties. Nevertheless, comparing average values highlights potential differences between emissions from Australian temperate forests and emissions from North American temperate forests. Emission factors for both hydrogen cyanide and ethene are in excellent

agreement, and emission factors for methanol, formaldehyde and 1,3-butadiene are within 20% of each other. Emission factors for ethane, acetylene and toluene also agree quite well, being within about 30% of each other. However, Australian forest fires potentially emit 50% more formic acid, twice as much acetic acid and ammonia, less than half as much ethanol and monoterpenes, and two to ten times more acetonitrile and pyrrole than North American fires. Lower emissions of compounds such as monoterpenes would impact downwind plume chemistry as the smoke is photochemically processed (Akagi et al.

2013). The use of Australian-specific emission factors is therefore recommended in studies looking at the regional impact of fires in Australian temperate forests.





**Table 5.** Comparison of VOC emission ratios and emission factors reported in the literature for fires in temperate forests in Australia and in North America. Emission ratios (ER) are in mol mol$^{-1}$ and emission factors (EF) are in g kg$^{-1}$ dry fuel burnt.

| Species | MW | ref. | This study — White cell FTIR and SIT-MS analysis of grab samples - prescribed fires in NSW ER | EF | Open-path FTIR - average values ER | EF | Lawson et al. 2015 ER | EF | Akagi et al. 2011 EF |
|---|---|---|---|---|---|---|---|---|---|
| Ammonia | 17 | CO | | | 0.023 (0.007) | 1.6 (0.6) | | | 0.8 (0.4) |
| Acetylene | 26 | CO$_2$ | 0.00037 ± 0.00008 | 0.35 ± 0.09 | | | | | 0.26 (0.04) |
| Hydrogen cyanide | 27 | CO | 0.0063 ± 0.0007 | 0.7 ± 0.2 | | | 0.0057 | 0.7 | 0.7 (0.2) |
| Ethene | 28 | CO | 0.009 ± 0.001 | 1.1 ± 0.2 | 0.011 (0.003) | 1.2 (0.2) | | | 1.2 (0.2) |
| Ethane | 30 | CO | 0.0038 ± 0.0003 | 0.48 ± 0.09 | 0.004 (0.001) | 0.5 (0.2) | 0.0032 | 0.41 | 0.6 (0.2) |
| Formaldehyde | 30 | CO | 0.018 ± 0.003 | 2.3 ± 0.5 | | 1.7 (0.4) | 0.011 | 1.6 | 2.1 (0.4) |
| Methanol | 32 | CO | 0.022 ± 0.002 | 3.0 ± 0.5 | 0.016 (0.005) | 2 (1) | 0.014 | 2.1 | 1.7 (0.5) |
| Acetonitrile | 41 | CO | 0.0038 ± 0.0005 | 0.7 ± 0.1 | | | 0.0013 | 0.25 | 0.12 (0.05) |
| Acetaldehyde | 44 | CO | 0.007 ± 0.001 | 1.3 ± 0.3 | | | 0.0044 | 0.92 | 0.8 (0.2) |
| Ethanol | 46 | CO | 0.00021 | 0.04 ± 0.01 | | | | | 0.10 (0.05) |
| Formic acid | 46 | CO | | | 0.003 (0.001) | 0.45 (0.16) | | | 0.29 (0.09) |
| Butadiene | 54 | CO$_2$ | 0.000074 ± 0.000009 | 0.23 ± 0.04 | | | | | 0.19 (0.05) |
| sum of acetone and propanal | 58 | CO | 0.0034 ± 0.0005 | 0.8 ± 0.2 | | | 0.002 | 0.54 | 0.54 (0.15) (acetone) 0.11 (0.05) (propanal) |
| Acetic acid | 60 | CO | | | 0.020 (0.009) | 4.5 (1.6) | | | 2.1 (0.7) |





| | | | This study | | | | References | | |
| | | | White cell FTIR and SIT-MS analysis of grab samples - prescribed fires in NSW | | Open-path FTIR - average values | | Lawson et al. 2015 | | Akagi et al. 2011 |
| Species | MW | ref. | ER | EF | ER | EF | ER | EF | EF |
|---|---|---|---|---|---|---|---|---|---|
| Pyrrole | 67 | CO | 0.0006 ± 0.0003 | 0.16 ± 0.08 | | | | | 0.012 (0.009) (pyrrole) 0.047 (0.026) (? MW67) |
| sum of furan and isoprene | 68 | CO | 0.0019 ± 0.0003 | 0.5 ± 0.1 | | | 0.0053 | 1.7 | 0.3 (0.1) (furan) 0.10(0.004) (isoprene) 0.18 (0.08) (? MW68) |
| sum of MACR, MVK and 2-butenal | 70 | CO | 0.0035 ± 0.0009 | 1.0 ± 0.3 | | | 0.0012 | 0.38 | 0.05 (0.02) (MACR) 0.16 (0.04) (MVK) 0.2 (0.1) (2-butenal) 0.3 (0.2) (? MW70) |
| Butanone | 72 | CO | 0.00082 ± 0.00007 | 0.25 ± 0.05 | | | 0.001 | 0.35 | 0.13 (0.04) (butanone) 0.09 (0.04) (? MW72) |
| Benzene | 78 | $CO_2$ | 0.00014 ± 0.00002 | 0.39 ± 0.07 | | | | 0.69 | 0.3 (0.1) |
| Toluene | 92 | CO | 0.0006 ± 0.0001 | 0.23 ± 0.05 | | | 0.00069 | 0.30 | 0.19 (0.05) |
| sum of $C_8H_{10}$ species | 106 | CO | 0.00025 ± 0.00005 | 0.11 ± 0.03 | | | 0.00053 | 0.26 | 0.17 (0.14) ($C_8$ aromatics) 0.2 (0.1) (benzaldehyde) |
| Monoterpenes | 136 | CO | 0.0009 ± 0.0002 | 0.5 ± 0.1 | | | 0.00018 | 0.11 | 0.9 (0.3) |





## 4.4 Comparison with emission factors reported for Australian savanna

As mentioned earlier, most of the area burnt in Australia annually is in the semi-arid and tropical savannas in the north of the country. A number of studies have characterised smoke from these fires (Hurst et al., 1994a, b, 1996; Shirai et al., 2003; Paton-Walsh et al., 2010; Meyer et al., 2012; Smith et al., 2014; Desservettaz et al., 2017; Wang et al., 2017a, b). Smith et al. (2014) used an OP-FTIR system to derive emission factors for $CO_2$, CO, $CH_4$, ethane, ethene, acetylene, formaldehyde, methanol, formic acid, acetic acid, ammonia and hydrogen cyanide. Comparing our OP-FTIR emission factors for temperate forests listed in Table 5 to those reported in Table 5 of Smith et al. (2014) indicates that both ecosystems have similar emission factors for formaldehyde and hydrogen cyanide (1.7 (0.4) vs. 1.6 (0.4) and 0.7 (0.2) vs 0.5 (0.3) g kg$^{-1}$ dry fuel burnt). Methane, methanol and ammonia show high variability in both ecosystems, and although the emission factors measured for temperate forests fires are higher, the emission factors agree within the uncertainties quoted (3.5 (0.9) vs. 2.2 (1.2), 2 (1) vs. 1.1 (0.8) and 1.6 (0.6) vs. 0.7 (0.4) g kg$^{-1}$ dry fuel burnt for methane, methanol and ammonia, respectively). The comparison also reveals that fires in Australian temperate forests emit up to five times more ethane, three times more acetic acid, formic acid and acetylene, and twice as much ethene than Australian savanna fires on a kg of dry fuel basis. This highlights the need for ecosystem-specific emission factors for Australia, especially when looking at regional impacts of biomass burning events.

## 5 Summary and Conclusions

In this study, emission factors were derived for a total of 25 trace gas species using a mixture of in situ open-path FTIR and grab sampling at nine prescribed fires in Australian temperate forests. MCE values measured during these ground-based measurements were not significantly different from those reported in the literature from airborne measurements, which contrasts with what has been observed in temperate ecosystems in North America. The emission factors for $CH_4$, ethene, formaldehyde, methanol, formic acid, acetic acid, the sum of furan and isoprene and the sum of acetone and propanal exhibited significant MCE dependence, although there were regional differences for formic acid, acetic acid and $CH_4$ that indicate that the use of MCE may be of limited use to extrapolate emission factors. There were also differences between the MCE dependences observed in this study compared to those observed for fires in North American temperate ecosystems.

The average emission factors measured for Australian temperate forest fires were compared to those measured for fires in North American temperate ecosystems. The average emission factors for hydrogen cyanide and ethene were in excellent agreement, and those of methanol, formaldehyde, ethane, toluene and 1,3-butadiene were in good agreement (within 30%). The emission factors measured in this study for other species however, indicate that Australian temperate forests may emit 50% more formic acid, twice as much acetic acid and ammonia, half as much ethanol and monoterpenes, and two to ten times more acetonitrile and pyrrole than North American fires on a per kg of dry fuel burnt basis.

We also find that the emission factors for hydrogen cyanide and formaldehyde for Australian temperate forest fires are in excellent agreement with those measured for Australian savanna fires, but that the forest fires have emission factors that are up to five times higher for ethane, three times higher for acetic acid, formic acid and acetylene, and twice higher for ethene.





These differences would impact plume chemistry and influence air quality outcomes downwind of the fires. We therefore recommend that the emission factors presented here and in other studies such as those of Lawson et al. (2015) and Paton-Walsh et al. (2014) be used in studies of biomass burning that require ecosystem-specific emission factors to represent emissions from
Australian forest fires.

*Data availability.* All the emission ratios and the emission factors measured as part of this study are summarized in .csv files provided as a supplement to the main text.

*Author contributions.* EAG contributed to field work in NSW, oversaw collection, instrumental analysis and data analysis for the grab samples, contributed to QA/QC of all data and wrote the manuscript. CPW conceived of the project, contributed to field work, oversaw
measurements, spectral analysis, data analysis and QA/QC for all open-path FTIR measurements. MJD deployed the open-path system at fires in Victoria and contributed to data analysis. TELS contributed to FTIR spectral analysis and MCE analysis. LV, CJW and CPM coordinated with the Department of Environment, Land, Water and Planning to make attendence at the fires in Victoria possible and contributed details of vegetation at the fires in Victoria. All authors contributed to manuscript editing.

*Competing interests.* The authors declare no competing interests.

*Acknowledgements.* For the NSW fires, the authors would like to acknowledge Sharon Evans, Bill Sullivan, and Simon Hawkes from the New South Wales National Parks and Wildlife Service, for allowing us to make measurements at their prescribed burns and providing copies of their burn plans. Thanks are also due to Melanie Cameron, Dagmar Kubistin, Paul Taglieri, and Rachel Stevens from the University of Wollongong and Grant Edwards and Cheryl Tang from Macquarie University for help with grab sample collection. For the fires in Victoria, we thank Elizabeth Ashman from the Department of Environment, Land, Water and Planning as well as Doreena Dominick and Kaitlyn
Lieschke from the University of Wollongong. We would also like to acknowledge Travis Naylor and David Griffith for helpful MALT discussions and Graham Kettlewell and Martin Riggenbach for technical support. This work was funded by the Australian Research Council as a small component of the Discovery Project DP110101948 (NSW fires) and as part of the Smoke Emission and Transport Modelling project commissioned and funded by the Department of Environment, Land, Water and Planning, Victoria. We also acknowledge the Clean Air and Urban Landscapes Hub of Australia's National Environmental Science Programme for funding the further analysis of the results that was required to produce this paper.



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
