# Peer review of "Emissions of trace gases from Australian temperate forest fires: emission factors and dependence on modified combustion efficiency"

_Atmospheric Chemistry and Physics, 2017_

## Referee Comment (RC1) · N. Surawski (Referee) · 6 Nov 2017

The reviewer thanks the authors for submitting their article entitled "Emissions of trace gases from Australian temperate forest fires: emission factors and dependence on modified combustion efficiency" to Atmospheric Chemistry and Physics for potential publication in this journal. In this article, the authors undertake trace gas measurements from nine prescribed fires in South-Eastern Australia (seven in NSW and two in Victoria). In this study, the main focus is on VOC measurements as well as trace gas emissions such as CO2, CO and CH4 that support analysis behind the measurements (i.e. they enable MCE to be calculated, for example). The authors use a combination of

open-path FTIR, SIFT-MS and White Cell spectroscopy as tools to quantify trace gas species. Within this field, state-of-the-art (at least from a North American perspective) has been advanced by authors such as R. J. Yokelson et al. and S. K. Akagi et al. which the authors of the current manuscript have cited. The opinion of the reviewer is that the current study presented by the authors is a timely addition to the literature. The authors demonstrate that ecosystem specific EFs should be used for VOC emissions accounting in Australia and also demonstrate that some VOC species differ significantly from those measured in North America. On these grounds I find the current contribution useful, and furthermore recommend publication after minor revisions and attending to some technical issues.

Whilst the reviewer is not an expert in VOC measurement and chemistry, Figure 3 demonstrates some nice results showing excellent agreement between ethene mixing ratios quantified with SIFT-MS and White cell FTIR spectroscopy. This gives the reviewer some confidence that the instrumentation used in this study is quantitatively reliable.

The manuscript is currently in fairly good shape; however, an accepted manuscript would have to attend to a few matters.

Title: The reviewer has a preference for titles that indicate the outcome. This would help to promote the findings of this paper and get more people to read it. Something in the title that indicates the recommendation of ecosystem-specific VOC emissions may help.

Abstract: Line 8. "... compare with Australian savanna". This is presumably due to a paucity of data in Australia. May help to indicate why comparison was done with a different biome.

Line 9. "... disagree by a factor of two or more". May help to indicate which VOCs differ by that amount.

Introduction: Line 15. "... carbon monoxide and aerosol". Probably reads better as ... particulate matter. Line 19. May help to be careful regarding the comment "... lower due to rapid regrowth". If there was a change in fire regime e.g. fire frequency then the rapid regrowth would not occur. Net C emissions would increase.

Line 15. "... pyro-convective lofting". I believe the authors have missed two papers here. Please consult:

de Laat, A. T. J., D. C. S. Zweers, R. Boers, and O. N. E. Tuinder (2012), A solar escalator: Observational evidence of the self-lifting of smokeand aerosols by absorption of solar radiation in the February 2009 Australian Black Saturday plume, J. Geophys. Res., 117, D04204,doi:10.1029/2011JD017016.

Siddaway, J. M., and S. V. Petelina (2011), Transport and evolution of the 2009 Australian Black Saturday bushfire smoke in the lowerstratosphere observed by OSIRIS on Odin, J. Geophys. Res., 116, D06203, doi:10.1029/2010JD015162

Line 23. "... weather conditions that are conducive to pollution build up". What conditions are these - a stable atmosphere? More detail required.

Line 15. Page 3. "... highly cited compilations". This phrase should not appear in a scientific article. It appears a bit like a sales job.

Methods. Line 30. Presumably bark litter was present too? Line 32. "... canopy species" then "... overstorey species". Choose one term and stick with it.

Section 2.2. Line 16. "... forest road ": looks like a firebreak to me.

Section 2.3.1. I'm not sure what the phrase "co-adding scans" means?

Section 2.3.2 Line 23. Is the dilution ratio measured or assumed? If measured, how was this done?

Figure 3. Can't see the vertical errors bars < 10 ppm. Have they been calculated? Also, how were these uncertainties calculated? This information should go into the

Figure caption.

Section 2.4. Line 8. Orthogonal regression. Please check this terminology. Greg Ayers refers to this as restricted major axis regression. It may help to cite this paper too - it's in Atmospheric Environment from memory.

Page 8. Lines 3-4. RE selection of reference species. Not really sure what the chemical reasoning is for these selections? Is this just a matter of choosing a reference species that correlates with your results, or is there some more fundamental reasoning sitting behind this?

Section 2.5 Page 8. Line 19. molar mass not molecular mass. Page 9. Line 18. "... uncertainties in quadrature". Are you able to shed more light on what this technique does?

Section 2.6. Line 20. The first part of this sentence mentions MCE then it moves to combustion efficiency. I think it should be the other way round? Define combustion efficiency and then define MCE as an approxmation.

Section 3.2. Page 13. Line 6-7. RE uncertainties. It may help to bring this information forward i.e. uncertainties calculated according to Paton-Walsh. The first mention of uncertainties in the methods may be a good locus for this information.

Section 3.3. Line 9. The p-value. Not sure what test was done to obtain this p-value?

Table 3. The table caption is not self-contained for the reader. How are the +/- values calculated? What test was done for the p-values? Putting some of this detail in the Table caption may help.

Section 4.1. Page 17. Lines 17-18. "... not only a product of combustion". Are there any quantitative insights regarding non-combustion emissions?

Section 4.2. Lines 29-30. "... wild or prescribed fires, or between measurement platforms". Just wondering whether you have a physical explanation for why wild versus

prescribed MCEs are similar.

Table 4. Last row. Smoke < 20 min. Not sure what this fire type relates too? Further details may help.

Section 4.3. Line 17. I think you mean Table 5?

Line 23. "... two to ten times more acetonitrile and pyrrole". Just a two-fold suggestion here. What is the role of these compounds in atmospheric chemistry and why, perhaps, you got the differences you did compared to Northern American fires.

Table 5. The last column needs to be tidied up a bit. There are question marks and undefined acronyms. Not sure what MACR and MVK relate to for example. Also, you have the use of MM (molecular mass) and MW (molecular weight) in the manuscript. Stick to one term.

Section 5. Page 23. Line 2. "... impacts plume chemistry". In what ways? Some discussion of these impacts in the discussion may round it out a bit more - at least in terms of impact.

Supplementary Information  Supporting Data. These aspects of the submission look satisfactory.

Thanks again to the authors for a timely submission regarding VOC emissions from temperate forest fires.  The reviewer wishes the authors good luck with the re-submission of this paper to ACP.

Reviewed by: N. Surawski, Sydney, Australia.

---

## Referee Comment (RC2) · V. Selimovic (Referee) · 14 Nov 2017

This reviewer would first like to thank the authors for submitting their article entitled "Emissions of trace gases from Australian temperate forest fires: emission factors and dependence on modified combustion efficiency" to Atmospheric Chemistry and Physics for potential publication in this journal, and finds the subject matter appropriate. In the article presented, the authors measure trace-gases and VOCs from nine prescribed fires, seven of which were in New South Wales and two in Victoria. Three different instruments were used in sampling, including an open-path FTIR, SIFT-MS, and White cell for grab sampling. While the results are relevant and attempt to close some gaps

in ecosystem specific emission factors, the manner in which the data are presented and manipulated needs work. The manuscript presented lacks critical information as to how sampling was conducted and how components were measured by different instrumentation. Additionally, the manipulation and presentation of the data collected suffers from inconsistencies that reduce the significance of the overall message the authors are attempting to present. This manuscript would benefit significantly from clarification as well as further discussion into how the data was analyzed, including justification for the methods used. Therefore, it is the opinion of this reviewer that this manuscript not be accepted until these issues are addressed in detail.

Abstract: The fires studied are prescribed fires and may not represent wildfires. See: Liu et al., 2017: "Airborne measurements of western U.S wildfire emissions: Comparison with prescribed burning and air quality implications"

Introduction

P1, L21-22: The following sentence "The mix of VOCs emitted during biomass burning may be ecosystem specific, especially for VOCs that are associated with biogenic processes (as opposed to combustion processes) and that are distilled from vegetation in the early stages..." is somewhat unclear. Are you arguing that fuel type can impact emissions? If so, this is true, but raising biogenic emissions here is confusing and the message could be clarified or omitted. For example, some biogenic compounds like monoterpenes are stored in plant tissue and can be emitted due to heat from a fire, but others (like isoprene) are made and emitted immediately. Therefore, isoprene is emitted all the time, but made in fires not by heating stored isoprene but breaking down solid biomass. Also, the concept of 'early stages' has no meaning in a moving landscape fire.

P2, L1: The OVOC are not distilled but are pyrolysis products instead.

P3, L17: Maybe include "we compare our results with the emission factors listed in Akagi et al...for temperature forests and to emission factors measured for Australian

savannah fires and find significant differences in both cases" in the abstract, with a quantitative comparison and list of differences for some compounds. You already sort of do it in the abstract, but elaborate a little more. I.E: "Some species agree within 20%, others differ by a factor of 2 or more." Which ones?

Methods

Sect 2.2, P4, "Open-path FTIR system": How do you measure pressure and temperature with the OP-FTIR? Looked to Paton-Walsh 2014 and didn't find anything explicit on how that was done. You mention temperature and pressure for the white cell later on, so having it for the OP-FTIR should be just as important.

Sect 2.3, P5, "Grab sampling": How were the glass grab samples filled? Was there a sample line? P6, L7: "As for the OP-FTIR spectra, mole fractions were retrieved using the Multiple Atmospheric Layer Transmission (MALT) model..." You already mention this in the OP-FTIR section. How were the spectra from the White cell analyzed? Were they also analyzed using MALT?

P6, L25: The authors mention mass to charge ratios and calibration factors used to quantify them in the supplemental. According to Table S2, H3O+ is used as a reagent ion for HCN and formaldehyde which were both additionally assigned the same sensitivity. Did the authors compare HCN and formaldehyde values to any other instruments, for instance, results from OP-FTIR or grab samples? HCN and formaldehyde both have proton affinities that aren't much higher than water, and sometimes this can be an issue, especially for instruments like a PTR-MS that use H3O+ as its reagent ion. Does SIFT-MS have similar issues? If so, they should be addressed with instrument comparisons. A figure like Figure 3 would be nice for compounds like HCN or formaldehyde.

P7, L15: "Also, not every trace gas species was present at a detectable level in every sample. For some fires, this resulted in too few samples to allow an emission ratio to be meaningfully derived by regression for that species. For this reason, emission

ratios for each species were also derived through combining samples from all fires."
Can you elaborate on this? The authors mention earlier in the paper that emissions
vary based on fuel type, so how can you justify combining samples from all fires? The
authors also mention further in the paper on P10 Line 17 that some species show
important site-to-site variability. In the supplement it looks like fuel types from the fires
were mostly dry sclerophyll, but the understory seemed to vary. Are you worried about
understory components contributing differences in ER? Selimovic et al., 2017 (currently
in ACP discussion) found that emissions vary based on fuel component, so this might
be something to consider reworking using a different method. The one presented in
Yokelson et al 2011, Figure 2 might be valid. Also, poor correlation or low sample
number is no reason not to report data, even a single sample is meaningful and should
be included.

P8, L1-4: This doesn't make sense mathematically. If benzene is not highly correlated
to CO or CO2, then that is real. If it has better correlation with ethene, it doesn't matter.
The uncertainty in benzene to ethene coupled with the uncertainty in ethene to CO or
CO2 should have the same overall uncertainty.

P9, L13: Using only three species in "CT" inflates the EF. It's easy enough to include
all C-containing gases and is also more accurate.

P10, L4: "These are indicative of the type of combustion (e.g flaming vs. smoldering)
captured by the grab sampling, and are not necessarily representative of the whole fire.
As an example, the average MCE of the grab samples collected at the Gulguer Plateau
fires was 0.78 +/- 0.09 whereas a fire-integrated value of 0.90 was measured by OP-
FTIR." Which MCE did the authors use in the data analysis stage? It is not explicitly
stated in the paper. For EF that were calculated using grab samples, was grab sample
MCE used or fire-integrated? 0.78 indicates a fire that is more smoldering, but 0.90
indicates a fire that is more flaming. This could be problematic when trying to make the
case for compounds emitted during the smoldering stage versus compounds emitted
during the flaming stage, especially in relation to MCE. It would be helpful if the data

analysis process was described in detail with all of the specifics. Results

P10, L17-32: I have issues with the authors choosing to exclude emissions of certain compounds from the Gulguer fire because it does not fit within the observed mean ratio without it, but then choosing to include emissions combined from all fires, despite site-to-site variability, which the Gulguer fire clearly shows. You should stick to one method or the other. Either include all of the samples regardless of how they affect the mean, or keep the ER fire-specific. Switching between the two reduces the significance of the message you're trying to get across. Additionally, fires are naturally variable and it's not representative to exclude data because of a low r-squared value. All samples without high instrumental error are valid and any number of samples from 1 to 'n' at some level of ecosystem specificity will give you your best results.

P11, Table1: Convert all of the ER to the same reference species for ease of use and eliminate the r-squared column, which isn't useful.

P13, L14: Within what % uncertainty? Be more quantitative.

P14, L5: What is meant by "fire-averaged?" Which fires?

P14, L8, Table 3: What p value? How was this calculated? Maybe include this in the table caption, or in detail in the paragraph.

P15, Fig 6. There is no inherent value in a high r-squared for EF vs. MCE. The r-squared is simply an indication of the dependence on flaming and smoldering and if other things like fuel chemistry or multiple formation mechanisms impact the EF vs. MCE then that is useful to see. The Lawson et al fire was in a heath land and seems less relevant that the Gulguer Plateau fire.

P16, L5: Why was methanol not included for the Gulguer Plateau fire? Nothing about this is mentioned earlier in the paper, and it's included as part of Table 2.

P16, L16-17: Burling et al. was spring fires and Akagi et al. sampled fires in the fall so a seasonal difference can contribute to the variability.

Discussion

P17, L8: Can you elaborate on why you think there is a relationship for the NSW but not when you include all fires? This seems to be further suggestion of site-to-site variability.

P17, L17: "that are biogenically produced by vegetation and are not only a product of combustion..." Please clarify. See comment 1 earlier, regarding a similar statement in the introduction.

P17, L28-29: "...relatively low average MCE of 0.91." Relatively low compared to what? Table 4 shows an even lower MCE of 0.89 for the same study.

P18, Table 4: Filling in the Akagi et al MCE based on the CO CO2 EF shown might make it easier to compare that aspect of studies quickly.

P19, L7: The results of the study should be included, even if the discussion isn't repeated. You should at least discuss how the comparison worked out.

P19, L10: The Lawson fire was not a temperate forest fire.

P19, L22-23: Do you think Nitrogen emissions higher due to seasonal high fuel N?

P23, L1: Can you elaborate on how they would impact plume chemistry and influence air quality outcomes downwind of the fires? Some discussion would be helpful.

Technical Corrections:

P1, L17: Change "At a national level, average gross annual emissions of total carbon from fires.." to "..annual emissions of total carbon from some fires.." since not all vegetation grows back fast.

P3, L3: You already mention Hurst et al. 1996 in page 2, line 31. You should remove the sentence from the third page and add it to the second, or vice versa. Either way I think consolidating the statements would be helpful, since having it in two locations essentially saying the same thing seems redundant.

P3, L6: Abbreviate New South Wales National Parks as NSW. You mention it in Page 3, Line 11, but don't abbreviate it before then.

P5, L5: "CO2, CO, CH4, acetic acid, ammonia, ethene,..." and "CO2, CO, CH4, ethane and ethene..." This could be considered "picky" but I think it would be useful to include the chemical formulas and names of all the compounds to maintain consistency (I.E: Carbon Dioxide (CO2), Carbon monoxide (CO), Methane (CH4), acetic Acid (CH3COOH), ammonia (NH3), etc).

P19, L17: "..emission factors listed in 5.." Do you mean Table 5?

Supplemental: No issues on the supplemental

References: Brilli, F., Gioli, B., Ciccioli, P., Zona, D., Loreto, F., Janssens, I. A., and Ceulemans, R.: Proton Transfer Reaction Time-of-Flight Mass Spectrometric (PTR-TOF-MS) determination of volatile organic compounds (VOCs) emitted from a biomass fire developed under stable nocturnal conditions, Atmospheric Environment, 97, 54-67, https://doi.org/10.1016/j.atmosenv.2014.08.007, 2014.

Liu, X., Huey, G. L., Yokelson, R. J., Selimovic, V., Simpson, I. J., Müller, M., Jimenez. J. L., Campuzano-Jost, P., Beyersdorf. A. J., Blake, D. R., Butterfield, Z., Choi, Y., Crounse, J. D., Day, D. A., Diskin, G. S., Dubey, M. K., Fortner, E., Hanisco, T. F., Hu, W., King, L. E., Kleinman, L., Meinardi, S., Mikoviny, T., Onasch, T. B., Palm, B. B., Peischl, J., Pollack, I. B., Ryerson, T. B., Sachse, G. W., Sedlacek, A. J., Shilling, J. E., Springston, S., St. Clair, J. M., Tanner, D. J, Peng, A. P., Wennberg, P. O., Wisthaler, A., and Wolfe, G. M.: Airborne measurements of western U.S wildfire emissions: Comparison with prescribed burning and air quality implications, J. Geophys. Res. Atmos., 122, 6108-6129, doi:10.1002/2016JD026315, 2017.

Koss, A. R., Sekimoto, K., Gilman, J. B., Selimovic, V., Coggon, M. M., Zarzana, K. J., Yuan, B., Lerner, B. M., Brown, S. S., Jimenez, J. L., Krechmer, J., Roberts, J. M., Warneke, C., Yokelson, R. J., and de Gouw, J.: Non-methane organic gas emissions

from biomass burning: identification, quantification, and emission factors from PTR-ToF during the FIREX 2016 laboratory experiment, Atmos. Chem. Phys. Discuss., https://doi.org/10.5194/acp-2017-924, in review, 2017.

Knighton, W. B., Fortner, E. C., Midey, A. J., Viggiano, A. A., Herndon, S. C., Wood, E. C., and Kolb, C. E.: HCN detection with a proton transfer mass reaction spectrometer, International Journal of Mass Spectrometery, 283, 112-121, https://doi.org/10.1016/j.ijms.2009.02.013, 2009. Selimovic, V., Yokelson, R. J., Warneke, C., Roberts, J. M., de Gouw, J., Reardon, J., and Griffith, D. W. T.: Aerosol optical properties and trace gas emissions by PAX and OP-FTIR for laboratory-simulated western US wildfires during FIREX, Atmos. Chem. Phys. Discuss., https://doi.org/10.5194/acp-2017-859, in review, 2017.

Yokelson, R. J., Burling, I. R., Urbanski, S. P., Atlas, E. L., Adachi, K., Buseck, P. R., Wiedinmyer, C., Akagi, S. K., Toohey, D. W., and Wold, C. E.: Trace gas and particle emissions from open biomass burning in Mexico, Atmos. Chem. Phys., 11, 6787-6808, https://doi.org/10.5194/acp-11-6787-2011, 2011.

---

## Author Comment (AC1) · 23 Jan 2018

The reviewer thanks the authors for submitting their article entitled "Emissions of trace gases from Australian temperate forest fires: emission factors and dependence on modified combustion efficiency" to Atmospheric Chemistry and Physics for potential publication in this journal. In this article, the authors undertake trace gas measurements from nine prescribed fires in South-Eastern Australia (seven in NSW and two in Victoria). In this study, the main focus is on VOC measurements as well as trace gas emissions such as $CO_2$, CO and $CH_4$ that support analysis behind the measurements (i.e. they enable MCE to be calculated, for example). The authors use a combination of open-path FTIR, SIFT-MS and White Cell spectroscopy as tools to quantify trace gas species. Within this field, state-of-the-art (at least from a North American perspective) has been advanced by authors such as R. J. Yokelson et al. and S. K. Akagi et al. which the authors of the current manuscript have cited. The opinion of the reviewer is that the current study presented by the authors is a timely addition to the literature. The authors demonstrate that ecosystem specific EFs should be used for VOC emissions accounting in Australia and also demonstrate that some VOC species differ significantly from those measured in North America. On these grounds I find the current contribution useful, and furthermore recommend publication after minor revisions and attending to some technical issues.

Whilst the reviewer is not an expert in VOC measurement and chemistry, Figure 3 demonstrates some nice results showing excellent agreement between ethene mixing ratios quantified with SIFT-MS and White cell FTIR spectroscopy. This gives the reviewer some confidence that the instrumentation used in this study is quantitatively reliable.
The manuscript is currently in fairly good shape; however, an accepted manuscript would have to attend to a few matters.

Title:
The reviewer has a preference for titles that indicate the outcome. This would help to promote the findings of this paper and get more people to read it. Something in the title that indicates the recommendation of ecosystem-specific VOC emissions may help.

As the ecosystem-specific emissions are only statistically significant for some of the gases, we feel that changing the title may be misleading and have decided to keep the original title.

Abstract:
Line 8. "... compare with Australian savanna". This is presumably due to a paucity of data in Australia. May help to indicate why comparison was done with a different biome.
Line 9. "... disagree by a factor of two or more". May help to indicate which VOCs differ by that amount.

We modified the abstract to address your comments and those of the other reviewer:

… We then compare our average emission factors to those measured for temperate forest fires elsewhere (North America) and for fires in another dominant Australian ecosystem (savanna) and find significant differences in both cases. Indeed, we find that although the emission factors of some species agree within 20%, including those of hydrogen cyanide, ethene, methanol, formaldehyde and 1,3-butadiene; others, such as acetic acid, ethanol, monoterpenes, ammonia, acetonitrile and pyrrole, differ by a factor of two or more.

Introduction:
Line 15. "... carbon monoxide and aerosol". Probably reads better as ... particulate matter.

The correction was made.

Line 19. May help to be careful regarding the comment "... lower due to rapid regrowth". If there was a change in fire regime e.g. fire frequency then the rapid regrowth would not occur. Net C emissions would increase.

The other reviewer also had misgivings about this paragraph. We removed "rapid" and added a reference.

Line 15. "... pyro-convective lofting". I believe the authors have missed two papers here. Please consult:
de Laat, A. T. J., D. C. S. Zweers, R. Boers, and O. N. E. Tuinder (2012), A solar escalator: Observational evidence of the self-lifting of smokeand aerosols by absorption of solar radiation in the February 2009 Australian Black Saturday plume, J. Geophys. Res., 117, D04204,doi:10.1029/2011JD017016.
Siddaway, J. M., and S. V. Petelina (2011), Transport and evolution of the 2009 Australian Black Saturday bushïnˇA¸re smoke in the lower stratosphere observed by OSIRIS on Odin, J. Geophys. Res., 116, D06203, doi:10.1029/2010JD015162

Thank you. We have added the suggested references.

Line 23. "... weather conditions that are conducive to pollution build up". What conditions are these - a stable atmosphere? More detail required.

We have added more detail in parentheses:
…under weather conditions (low wind speeds, stable atmosphere) that are conducive to pollution build up…

Line 15. Page 3. "... highly cited compilations". This phrase should not appear in a scientific article. It appears a bit like a sales job.

We only meant that the emission factors contained in this compilation are widely used (and do not include any results from Australian forest fires). The sentence was rephrased:

Currently, widely used compilations of emission factors …

Methods.
Line 30. Presumably bark litter was present too?

Yes, although it was not a dominant component of the ground litter. We added it to the list:

The ground cover was generally made up of native grasses and a litter of eucalypt leaves, bark and twigs, as well as fallen tree limbs of varying sizes.

Line 32. "... canopy species" then "... overstorey species". Choose one term and stick with it.

The two sentences were merged to remove the word "canopy":

In Victoria, dominant overstorey species were …

Section 2.2. Line 16. "... forest road ": looks like a firebreak to me.

Both "forest road" and "forest track" have been replaced by "fire trail" in the text, which is the term used on the maps provided by fire personnel.

Section 2.3.1. I'm not sure what the phrase "co-adding scans" means?

This sentence has been rewritten to clarify the meaning:

A spectrum consisting of 78 scans was acquired for each grab sample.

Section 2.3.2 Line 23. Is the dilution ratio measured or assumed? If measured, how was this done?

The dilution ratio was not measured, but is estimated from the temperature, pressure and flows of sample and carrier gases.

The flow tube dilution ratio under standard operating conditions is about 1:15.

Figure 3. Can't see the vertical errors bars < 10 ppm. Have they been calculated?
Also, how were these uncertainties calculated? This information should go into the Figure caption.

The vertical error bars are small, hence difficult to see. The revised version has a slightly larger figure, and was plotted without data symbols/dots, which makes the error bars more visible. The error bars on the y-axis are the standard deviation of the mean of the 8 measurements made by the SIFT-MS. On the x-axis, the error bars are the error on the fit reported by MALT. This information has been added to the text in the appropriate sections:

The uncertainty on individual grab sample measurements is the error on the retrieval reported by MALT. (Sect 2.3.1)

The standard deviation of the mean was taken as the uncertainty on the average mole fraction. (Sect. 2.3.2)

 and to the caption:

Error bars for the SIFT-MS are the standard deviation of the measurement, for the White cell FTIR they are the error on the retrieval.

Section 2.4. Line 8. Orthogonal regression. Please check this terminology. Greg Ayers refers to this as restricted major axis regression. It may help to cite this paper too - it's in Atmospheric Environment from memory.

Restricted major axis regression (RMA), major axis regression (MA) and orthogonal regression (also known as Deming regression) are all variants of "error-in-variable" regression models – they all minimize the deviations from the line of fit in both the x and y axes. They differ in the assumptions made about the error ratio $\lambda$ – the ratio of the total error variances of X and Y. RMA assumes $\lambda = s_x/s_y$, MA assumes $\lambda = 1$ and orthogonal regression lets you specify the weights $\omega$ for each data point individually, with $\lambda = \omega_x/\omega_y$, where $\omega_{xi} = 1/\sigma_{xi}^2$.

In this study, we used orthogonal regression because it let us take into account the uncertainty of individual measurements. This means that the line of best fit has greater dependence on the more precise data points. This is also the type of regression that gave the best results in a recent comparison of several regression techniques for application in atmospheric science (Wu and Yu, 2017).

Wu and Yu (2017) also noted that the effect of which regression method is used on the resulting slope is minimized for data that are highly correlated.

The following has been added to the text:

The regression is also weighted by the uncertainties in both x and y, which, in this case, are the measurement uncertainties as described above, so that the line of best fit has greater dependence on the more precise data points. As noted in a recent evaluation of linear regression techniques (Wu and Yu 2017), the type of linear regression applied has little impact on the resulting slope as long as the correlation coefficient is high. For this reason, we chose pairs of species that were best correlated to derive emission ratios and do not report results when $R^2 < 0.5$, as this should yield the most robust results.

Page 8. Lines 3-4. RE selection of reference species. Not really sure what the chemical reasoning is for these selections? Is this just a matter of choosing a reference species that correlates with your results, or is there some more fundamental reasoning sitting behind this?

We chose species that correlate well with each other as this should lead to more robust slope values (as explained above); however, strong correlation may also have a physical meaning, indicating that the species are co-emitted. This has been added to the text at the end of Sect 2.4:

Good correlation between VOC species may indicate co-emission.

Section 2.5 Page 8. Line 19. molar mass not molecular mass.

The correction was made.

Page 9. Line 18. "...uncertainties in quadrature". Are you able to shed more light on what this technique does?

This is a standard error propagation technique. Assuming that errors are not correlated, then they can be added in quadrature: $\sigma z^2 = z^2 *[(\sigma_x/x)^2 + \sigma_y/y)^2]$

Section 2.6. Line 20. The first part of this sentence mentions MCE then it moves to combustion efficiency. I think it should be the other way round? Define combustion efficiency and then define MCE as an approximation.

Both have already been defined in the introduction, in the order suggested by the reviewer. In this section, we are introducing the equations used in the analysis (MCE).

Section 3.2. Page 13. Line 6-7. RE uncertainties. It may help to bring this information forward i.e. uncertainties calculated according to Paton-Walsh. The first mention of uncertainties in the methods may be a good locus for this information.

Agreed. We have moved this to the methods:

The uncertainty on individual measurements is the error on the retrieval reported by MALT. For a complete uncertainty budget on the OP-FTIR smoke measurements in smoke, see Appendix B in Paton-Walsh et al. (2014).

Section 3.3. Line 9. The p-value. Not sure what test was done to obtain this p-value?

The p-value is the probability that the slope is in fact zero (null hypothesis). This is one of the standard diagnostics of linear regression. We added the following to the text:

The strength of the relationship is judged from the coefficient of determination ($R^2$) and the p-value (the probability that there is no correlation between x and y).

Table 3. The table caption is not self-contained for the reader. How are the +/- values calculated? What test was done for the p-values? Putting some of this detail in the Table caption may help.

The caption now reads:

Summary of regression statistics for the emission factor dependence on modified combustion efficiency (MCE) of carbon-containing species measured by open-path FTIR in temperate forest fires in Australia

Section 4.1. Page 17. Lines 17-18. "... not only a product of combustion". Are there any quantitative insights regarding non-combustion emissions?

The sentence was removed from the text in response to comments from the other reviewer.

Section 4.2. Lines 29-30. "... wild or prescribed fires, or between measurement platforms". Just wondering whether you have a physical explanation for why wild versus prescribed MCEs are similar.

No, we do not have a physical explanation. A similar MCE indicates that a similar mix of flaming and smouldering combustion was captured. This could be coincidental, or an artefact of the sampling approaches, or could mean that the fires sampled did burn at a similar MCE. There is not enough data to draw definite conclusions. Even if the fires did burn at a similar MCE, this does not guarantee that their emissions would be the same: as the other reviewer pointed out, Liu et al. (2017) saw higher PM emission from wildfires than for prescribed fires burning at the same MCE. The following was added to the MCE discussion in Section 4.2:

The good agreement for MCE between platforms and fire type could be coincidental, or an artefact of the sampling approaches, or may in fact indicate that the prescribed and wild fires sampled burnt at a similar MCE. Liu et al. (2017) report EF for PM that are a factor of two higher for wildfires than for prescribed fires burning at the same MCE but do not observe the same for trace gases such as $CH_4$.

Table 4. Last row. Smoke < 20 min. Not sure what this fire type relates too? Further details may help.

The Akagi compilation only includes studies that sampled smoke that was less than 20 minutes old. However, we agree that this was not helpful, and have replaced this by "Prescribed & wild fires" since these are the types of fires sampled in the studies included in the compilation.

Section 4.3. Line 17. I think you mean Table 5?

Yes, thank you. The correction has been made.

Line 23. "... two to ten times more acetonitrile and pyrrole". Just a two-fold suggestion here. What is the role of these compounds in atmospheric chemistry and why, perhaps, you got the differences you did compared to Northern American fires.

Acetonitrile is somewhat long-lived in the atmosphere (months) and is considered a tracer for transported biomass plumes whereas more reactive nitrogen-containing species such as pyrrole may be tracers for fresher plumes. The observed variability in emission factors would be an important factor when interpreting plume age.

The difference with the North American fires may be due to differences in fuel nitrogen content. Most studies do not report fuel composition (this one included) so it is difficult to draw conclusions. Acacias are nitrogen-fixing and tend to have higher nitrogen content in their leaves than many other trees, which translates to higher N in the leaf litter as well (Snowdon et al., 2005). Acacias were present in the understorey of many of the fires sampled in this study, so this may be a contributing factor.
The following has been added to the discussion:

Nitrogen-containing VOCs make little contribution to the overall reactivity of a smoke plume (Gilman et al., 2015). Acetonitrile has an atmospheric lifetime on the order of months  and is a tracer for long-range transport of biomass plumes (Bange and Williams, 2000) whereas more reactive nitrogen containing-species may be tracers for fresh plumes (Gilman et al., 2015, Coggon et al., 2016). Higher emissions may affect estimates of plume age based on these species.
The difference with the North American fires may be due to higher fuel nitrogen content. *Acacia* are nitrogen-fixing species that have high leaf N content (1.50-3.55%) which is partly conserved through leaf fall, leading to higher nitrogen in the leaf litter (Snowdon et al., 2005). Acacia are some of the dominant understorey species in the forests investigated in this study, and their presence may have contributed to the high emissions of nitrogen-containing species; however, without fuel composition measurements, it is impossible to draw definitive conclusions.

Table 5. The last column needs to be tidied up a bit. There are question marks and undefined acronyms. Not sure what MACR and MVK relate to for example.

We have replaced the abbreviations by the full names of the molecules (MACR = methacrolein, MCK = methyl vinyl ketone). The abbreviations were also added in the text. The question marks represented compounds that were detected but unidentified in the reference. The question marks were removed and the caption modified:

Unidentified species that are likely to contribute to the signal measured by SIFT-MS are listed by their molar mass in the last column.

Also, you have the use of MM (molecular mass) and MW (molecular weight) in the manuscript. Stick to one term.

We have replaced MW by MM in Table 5 so that it is used consistently throughout the manuscript.

Section 5. Page 23. Line 2. "... impacts plume chemistry". In what ways? Some discussion of these impacts in the discussion may round it out a bit more - at least in terms of impact.

The sentence was modified to be more general (instead of focused on monoterpenes):

The initial mixture of trace gases emitted by a fire is one of the factors (along with meteorology and the presence of other sources) that influences plume aging (Akagi et al., 2012, Jaffe and Wigder, 2012) and therefore air quality outcomes downwind of the fires.

Supplementary Information Supporting Data. These aspects of the submission look satisfactory.

Thanks again to the authors for a timely submission regarding VOC emissions from temperate forest fires. The reviewer wishes the authors good luck with the resubmission of this paper to ACP.

Reviewed by: N. Surawski, Sydney, Australia.
This reviewer would first like to thank the authors for submitting their article entitled "Emissions of trace gases from Australian temperate forest fires: emission factors and dependence on modified combustion efficiency" to Atmospheric Chemistry and Physics for potential publication in this journal, and finds the subject matter appropriate. In the article presented, the authors measure trace-gases and VOCs from nine prescribed fires, seven of which were in New South Wales and two in Victoria. Three different instruments were used in sampling, including an open-path FTIR, SIFT-MS, and White cell for grab sampling. While the results are relevant and attempt to close some gaps in ecosystem specific emission factors, the manner in which the data are presented and manipulated needs work. The manuscript presented lacks critical information as to how sampling was conducted and how components were measured by different instrumentation. Additionally, the manipulation and presentation of the data collected suffers from inconsistencies that reduce the significance of the overall message the authors are attempting to present. This manuscript would benefit significantly from clarification as well as further discussion into how the data was analyzed, including justification for the methods used. Therefore, it is the opinion of this reviewer that this manuscript not be accepted until these issues are addressed in detail.

Abstract: The fires studied are prescribed fires and may not represent wildfires. See:
Liu et al., 2017: "Airborne measurements of western U.S wildfire emissions: Comparison with prescribed burning and air quality implications"

Thank you. I have added this reference to the discussion of MCE measured from different platforms for prescribed and wild fires (Section 4.2).

Introduction
P1, L21-22: The following sentence "The mix of VOCs emitted during biomass burning may be ecosystem specific, especially for VOCs that are associated with biogenic processes (as opposed to combustion processes) and that are distilled from vegetation in the early stages..." is somewhat unclear. Are you arguing that fuel type can impact emissions? If so, this is true, but raising biogenic emissions here is confusing and the message could be clarified or omitted. For example, some biogenic compounds like monoterpenes are stored in plant tissue and can be emitted due to heat from a fire, but others (like isoprene) are made and emitted immediately. Therefore, isoprene is emitted all the time, but made in fires not by heating stored isoprene but breaking down solid biomass. Also, the concept of 'early stages' has no meaning in a moving landscape fire.

The sentence has been modified:
The mix of VOCs emitted during biomass burning may be ecosystem-specific, with species such as monoterpenes being distilled from the vegetation as it is heated by the approaching fire …

P2, L1: The OVOC are not distilled but are pyrolysis products instead.

The mechanism of release is not always specified in the studies cited, so we use the general term 'heated'.

P3, L17: Maybe include "we compare our results with the emission factors listed in Akagi et al … for temperature forests and to emission factors measured for Australian savannah fires and find significant differences in both cases" in the abstract, with a quantitative comparison and list of differences for some compounds. You already sort of do it in the abstract, but elaborate a little more. I.E: "Some species agree within 20%, others differ by a factor of 2 or more." Which ones?

We did not add the reference to Akagi et al., (2011), as referencing is not recommended in the abstract, but we have modified the abstract as per your recommendations and those of the other reviewer:

We then compare our average emission factors to those measured for temperate forest fires elsewhere (North America) and for fires in another dominant Australian ecosystem (savanna) and find significant differences in both cases. Indeed, we find that although the emission factors of some species agree within 20%, including those of hydrogen cyanide, ethene, methanol, formaldehyde and 1,3-butadiene; others, such as acetic acid, ethanol, monoterpenes, ammonia, acetonitrile and pyrrole, differ by a factor of two or more.

Methods
Sect 2.2, P4, "Open-path FTIR system": How do you measure pressure and temperature with the OP-FTIR? Looked to Paton-Walsh 2014 and didn't find anything explicit on how that was done. You mention temperature and pressure for the white cell later on, so having it for the OP-FTIR should be just as important.

Thanks for pointing this out. The temperature and pressure are measured with sensors located near the spectrometer. The following has been added to the text:

Ambient pressure and temperature are monitored at one end of the path, through a barometer (Vaisala PTB110) and a resistance temperature detector (RTD PT100) connected to the computer controlling the spectrometer via an I/O box. The output is logged at the same time resolution as the spectral measurements.

Sect 2.3, P5, "Grab sampling": How were the glass grab samples filled? Was there a sample line?

No. This has been clarified in the text:

Samples were collected in 600 ml glass flasks, except at the Gulguer Plateau fire, where samples were collected into 1 L Tedlar bags. The glass flasks were pre-evacuated using a turbo-molecular pump (Pfeiffer TCS 010) prior to deployment to the fires, and filled with smoke on site by opening them for a few seconds. **No sample line was affixed to the flasks for sampling; flasks were positioned in the smoke prior to opening them.** The bags were flushed with high purity nitrogen and brought to the Gulguer Plateau fire where they were filled with smoke using a differential pressure system or 'vacuum box' powered by a generator. **As the generator had to be placed away from the fire, a sample line (~5 meters) was attached to the vacuum box**. Filling the bags took a

few minutes, and consequently, most samples were collected from large smouldering targets after the fire front had moved through the sampling area.

P6, L7: "As for the OP-FTIR spectra, mole fractions were retrieved using the Multiple Atmospheric Layer Transmission (MALT) model…" You already mention this in the OP-FTIR section. How were the spectra from the White cell analyzed? Were they also analyzed using MALT?

Yes, they were analyzed using MALT. The sentence was unclear and has been modified:

Mole fractions were retrieved using the Multiple Atmospheric Layer Transmission (MALT) model …

P6, L25: The authors mention mass to charge ratios and calibration factors used to quantify them in the supplemental. According to Table S2, $H_3O^+$ is used as a reagent ion for HCN and formaldehyde which were both additionally assigned the same sensitivity. Did the authors compare HCN and formaldehyde values to any other instruments, for instance, results from OP-FTIR or grab samples? HCN and formaldehyde both have proton affinities that aren't much higher than water, and sometimes this can be an issue, especially for instruments like a PTR-MS that use $H_3O^+$ as its reagent ion. Does SIFT-MS have similar issues? If so, they should be addressed with instrument comparisons. A figure like Figure 3 would be nice for compounds like HCN or formaldehyde.

The OP-FTIR system is fitted with a mercury cadmium telluride (MCT) detector, which does not allow us to measure HCN in the open-path. The OP-FTIR does measure formaldehyde however, and the results from the grab samples (SIFT-MS) are in good agreement (average of 2.3 vs 1.7 g/kg fuel) considering that the instruments sampled different fires.

It was not possible to determine formaldehyde by FTIR in the grab samples directly, since the in situ system is equipped with an indium antimony (InSb) detector which does not cover the appropriate spectral range. In fact, the only species that was measured by all three instruments was ethene.

It is true that both formaldehyde and HCN have proton affinities close to that of water and this can cause issues such as low sensitivity and dependence on water density in PTR measurements. However, these issues are much reduced in SIFT-MS measurements. Španěl et al. (1999) find that the [H3CO+]/[H3O+] ratio does not depend on [H2O]. Similarly,  Španěl et al. (2004) find that:

"…the ratio of the count rate of $H_2CN^+$ to that of $H_3O^+$ does not change dramatically with the $H_2O$ number density in the carrier gas, [$H_2O$]. Actually, the change is only 20% within the range of water concentration covered by these measurements (ranging from those typical of laboratory air and exhaled breath samples, i.e. relative humidity range 1–6%)."

Since HCN and formaldehyde have similar m/z (and are therefore likely to be transmitted in a similar way through the instrument), similar proton affinities, similar kinetics and little water dependence when measured by SIFT-MS, it seems reasonable to estimate the sensitivity of HCN from that of formaldehyde.

The following has been added to the supplementary ():

It should be noted that hydrogen cyanide was assigned the same calibration factor as formaldehyde. Both species have a similar m/z (and are therefore likely to be transmitted in a similar way through the instrument), similar proton affinities, similar kinetics and little water dependence when measured by SIFT-MS (Španěl et al., 1999, Španěl et al., 2004). Similarly, pyrrole was assigned the same calibration factor as isoprene.

P7, L15: "Also, not every trace gas species was present at a detectable level in every sample. For some fires, this resulted in too few samples to allow an emission ratio to be meaningfully derived by regression for that species. For this reason, emission ratios for each species were also derived through combining samples from all fires."
Can you elaborate on this? The authors mention earlier in the paper that emissions vary based on fuel type, so how can you justify combining samples from all fires? The authors also mention further in the paper on P10 Line 17 that some species show important site-to-site variability. In the supplement it looks like fuel types from the fires were mostly dry sclerophyll, but the understory seemed to vary. Are you worried about understory components contributing differences in ER? Selimovic et al., 2017 (currently in ACP discussion) found that emissions vary based on fuel component, so this might be something to consider reworking using a different method. The one presented in Yokelson et al., 2011, Figure 2 might be valid. Also, poor correlation or low sample number is no reason not to report data, even a single sample is meaningful and should be included.

A single sample is meaningful only if background values are known, which is not the case in our study. Most of the background values were below the limit of detection of the SIFT-MS. Therefore we chose to derive ER using linear regression. This has been clarified in the text:

More generally, we chose to use linear regression to derive ER instead of calculating a value from each measurement (e.g. Burling et al., 2011) because the background mole fractions of many measured species were poorly defined, often being below the detection limit of the SIFT-MS. Deriving emission ratio through regression without first subtracting background values introduces very little error (< 0.1%, Wooster et al., 2011).

We combined data from all fires to get an "ecosystem" ER, that hopefully captures variability due to different fuel types. These "ecosystem" ER agree well with the average ER calculated from the individual fires. We have added this clarification in the text:

Emission ratios were derived from the open-path measurements for each fire separately. The mean ER from all the fires sampled is then our best estimate for the ecosystem.
For the grab samples, emission ratios were derived for individual fires when possible; however, the VOC results from the targeted grab sampling were more highly variable than the open-path measurements in the well-mixed smoke, as is common for this type of sampling (Yokelson et al., 2008, 2013; Burling et al., 2011; Akagi et al., 2013). This resulted in poor correlations ($R^2 < 0.5$) for some species for certain fires. Also, not every trace gas species was present at a detectable level in every sample. For some fires, this resulted in too few samples to allow an emission ratio to be meaningfully derived by regression for that species. As ER were not successfully derived for each fire for some species, a mean ER was not necessarily the best estimate for the ecosystem. To derive a best estimate for the ecosystem, all valid samples were combined irrespective of which fire they were collected at and a single ER derived through orthogonal regression.

P8, L1-4: This doesn't make sense mathematically. If benzene is not highly correlated to CO or $CO_2$, then that is real. If it has better correlation with ethene, it doesn't matter. The uncertainty in benzene to ethene coupled with the uncertainty in ethene to CO or $CO_2$ should have the same overall uncertainty.

This method was not chosen to reduce uncertainties as such. We chose to only use pairs of well correlated trace gases as this reduces the impact of which regression method is used. There are several ways to handle data that has error in both the y and x axes, and although we chose a regression method that gives reliable results, it is noted that more robust slopes are obtained at higher correlation coefficients, i.e. the same slope is obtained whatever the regression method used at very high $R^2$ values, but differences between methods become more apparent as $R^2$ decreases (Wu and Yu, 2017).

P9, L13: Using only three species in "CT" inflates the EF. It's easy enough to include all C-containing gases and is also more accurate.

Although many carbon-containing gases were measured in the grab samples, only $CO_2$, CO and $CH_4$ were measured successfully in all samples. So that we can be consistent in the way we calculate EF from the grab samples, we therefore only use these three species in our calculations. It is true that this inflates the EF slightly (a few percent). We have added the following to the text:

For this analysis, emission factors for $CO_2$, CO and $CH_4$ were calculated for each individual grab sample using Eq. 4, with $C_T$ calculated as the sum of $CO_2$, CO and $CH_4$ only. Although many more carbon-containing species were measured in the grab samples, only $CO_2$, CO and $CH_4$ were successfully quantified in every single grab sample. For consistency, they were therefore the only species included in the calculation. Doing so inflates the emission factors by up to a few percent (< 5 %).

P10, L4: "These are indicative of the type of combustion (e.g flaming vs. smoldering) captured by the grab sampling, and are not necessarily representative of the whole fire. As an example, the average MCE of the grab samples collected at the Gulguer Plateau fires was 0.78 ± 0.09 whereas a fire-integrated value of 0.90 was measured by OPFTIR." Which MCE did the authors use in the data analysis stage? It is not explicitly stated in the paper. For EF that were calculated using grab samples, was grab sample MCE used or fire-integrated? 0.78 indicates a fire that is more smoldering, but 0.90 indicates a fire that is more flaming. This could be problematic when trying to make the case for compounds emitted during the smoldering stage versus compounds emitted during the flaming stage, especially in relation to MCE. It would be helpful if the data analysis process was described in detail with all of the specifics.

We have clarified what analysis was done for each type of measurements in the methods. To derive EF from the grab samples, we used the OP-FTIR CO and CO2 EF from the NSW fires, as mentioned in Sect 2.5:

Similarly, the MCE of a fire sampled by OP-FTIR was determined from the total excess amounts of $CO_2$ and CO detected by the open-path system (i.e. by summing the excess amounts from each measurement recorded). These MCE values are used to determine whether the emission factors of the species measured by OP-FTIR have a dependence on MCE.

For grab samples, two variants of the analysis were completed. The first one was used to derive emission factors and MCE values to evaluate whether the emission factors of the species measured only in the grab samples have a dependence on MCE.

Results
P10, L17-32: I have issues with the authors choosing to exclude emissions of certain compounds from the Gulguer fire because it does not fit within the observed mean ratio without it, but then choosing to include emissions combined from all fires, despite site-to-site variability, which the

Gulguer fire clearly shows. You should stick to one method or the other. Either include all of the samples regardless of how they affect the mean, or keep the ER fire-specific. Switching between the two reduces the significance of the message you're trying to get across. Additionally, fires are naturally variable and it's not representative to exclude data because of a low r-squared value. All samples without high instrumental error are valid and any number of samples from 1 to 'n' at some level of ecosystem specificity will give you your best results.

Agreed. The mean emission ratio in Table 1 includes the Gulguer Plateau fire. This mean value is not significantly different from the ER derived by combining the data from the four other fires.

Similarly, the emission ratio of acetonitrile to CO is markedly lower at Gulguer Plateau than at the other fires. This could be due to the lower nitrogen content of logs compared to foliage and twigs (Susott et al., 1996, Snowdon et al., 2005), resulting in lower emissions of nitrogen-containing species (Coggon et al., 2016).
The Gulguer Plateau fire is excluded from the emission ratio for acetonitrile derived from combining data from all fires, since including it results in $R^2 < 0.5$. Figure 5 shows the correlations of acetonitrile with CO; the Gulguer Plateau fire is shown in red, the other four fires are shown in black. The emission ratio derived from the black line is not significantly different from the mean ER that includes the Gulguer Plateau data (see Table 1).

As for results with low $R^2$, see our comments above concerning the robustness of slopes derived from poorly correlated pairs of species.

P11, Table1: Convert all of the ER to the same reference species for ease of use and eliminate the r-squared column, which isn't useful.

ER to CO (or $CO_2$) for all species are listed in Table 5. Table 1 is meant to be a transparent summary of how the grab sample data was processed, and as such we would like to keep Table 1 as it is now. And for reasons stated above, we believe that $R^2$ is a useful metric, indicating how robust the ER are likely to be.

P13, L14: Within what % uncertainty? Be more quantitative.

The uncertainties are listed in Table 2. We have clarified the text:

The differences are slight however, and the emission factors from Victoria agree within the uncertainties with those from NSW.

P14, L5: What is meant by "fire-averaged?" Which fires?

The fires measured by OP-FTIR. We have removed "fire-averaged" from the sentence.

P14, L8, Table 3: What p value? How was this calculated? Maybe include this in the table caption, or in detail in the paragraph.

The p-value is the probability that there is in fact zero no correlation between x and y (null hypothesis). This is one of the standard diagnostics of linear regression. We added the following to the text:

The strength of the relationship is judged from the coefficient of determination ($R^2$) and the p-value (the probability that there is no correlation between x and y).

P15, Fig 6. There is no inherent value in a high r-squared for EF vs. MCE. The r-squared is simply an indication of the dependence on flaming and smoldering and if other things like fuel chemistry or multiple formation mechanisms impact the EF vs. MCE then that is useful to see. The Lawson et al fire was in a heath land and seems less relevant that the Gulguer Plateau fire.

We report $R^2$ exactly for the reasons mentioned by the reviewer – as a means a judging to what extent MCE explains the variability observed in the EF. We have added this to the text:

A poor $R^2$ indicates that MCE alone cannot explain the variability in EF.

The OP-FTIR results from the Gulguer Plateau fire are included in Figure 6, only the grab sample results from this fire are not shown. For methanol, this is because no value is available (Table 1 shows the average of the 4 other fires, I have added a note to the table). For methane, there is a value, but the average MCE is 0.78 and falls outside the range measured by OP-FTIR. We have clarified the text:

Figure 6 also shows the average results derived for $CH_4$ and methanol from the grab samples. The grab sampling results from the Gulguer Plateau fire are either not available (methanol) or fall outside the range measured by OP-FTIR (methane) and therefore do not appear in Fig. 6.

and the caption for Figure 6:

The black circles represent average results from grab samples at four fires (The grab sampling results from the Gulguer Plateau fire are either not available (methanol) or fall outside the range measured by OP-FTIR (methane) and therefore do not appear).

P16, L5: Why was methanol not included for the Gulguer Plateau fire? Nothing about this is mentioned earlier in the paper, and it's included as part of Table 2.

No ER could be determined for methanol for the Gulguer fire in the grab samples. This is now indicated by a note in Table 1. The OP-FTIR data from that fire is included in Figure 6 (and the MCE analysis).

Discussion
P16, L16-17: Burling et al. was spring fires and Akagi et al. sampled fires in the fall so a seasonal difference can contribute to the variability.
P17, L8: Can you elaborate on why you think there is a relationship for the NSW but not when you include all fires? This seems to be further suggestion of site-to-site variability.

It does suggest site-to-site variability. This was touched on earlier, in Sect. 3.2: "This indicates a difference in emissions from the different regions sampled that is not explained by the difference in modified combustion efficiency". We have added the following to the discussion:

Considering the variability of relationships to MCE observed even for similar ecosystems, it seems likely that other factors are influencing emissions. Burling et al. (2011) sampled spring fires whereas Akagi et al. (2013) sampled autumn fires so it is possible that some of the variability is due to seasonal differences. In this study, fires were sampled over several years, both in spring (August-

September) and in autumn (April-May). There is no obvious seasonal effect in the data, however there seems to be regional effects, especially for formic acid and acetic acid, and these may be due to differences in vegetation.

P17, L17: "that are biogenically produced by vegetation and are not only a product of combustion …" Please clarify. See comment 1 earlier, regarding a similar statement in the introduction.

The sentence was removed.

P17, L28-29: "… relatively low average MCE of 0.91." Relatively low compared to what? Table 4 shows an even lower MCE of 0.89 for the same study.

Low for a very large wildfire. The sentence was removed.

P18, Table 4: Filling in the Akagi et al MCE based on the CO CO2 EF shown might make it easier to compare that aspect of studies quickly.

A value of ~0.92 was estimated and added to Table 4.

P19, L7: The results of the study should be included, even if the discussion isn't repeated. You should at least discuss how the comparison worked out.

The following sentence has been added:

They found good agreement for methanol and formaldehyde, and evidence for depletion of ammonia and ethene and formation of formic acid in aged smoke.

P19, L10: The Lawson fire was not a temperate forest fire.

True. However, the vegetation that burned on Robbins Island was similar in structure to the understorey that burnt in the prescribed fires we sampled. It is also the only other study in Australia to have calculated emission factors for a large number of species, so the comparison is still useful. This has been clarified in the text:

The plume was advected to the Station from a fire in coastal heath on a nearby island, mostly at night (from 23:00 AEST until 09:00 AEST). The vegetation burnt in the Robbins Island fire is similar to what typically burns in a prescribed fire, so their emission ratios and emission factors for VOCs are listed alongside ours in Table 5.

P19, L22-23: Do you think Nitrogen emissions higher due to seasonal high fuel N?

We sampled fires both in spring (Aug-Sep) and autumn (Apr-May), and see no obvious differences in emissions between the seasons. We do not have fuel composition data so it is difficult to draw conclusions.  One potential factor is the presence of Acacia species in the understorey, which are nitrogen-fixing:

The difference with the North American fires may be due to higher fuel nitrogen content. *Acacia* are nitrogen-fixing species that have high leaf N content (1.50-3.55%) which is partly conserved through leaf fall, leading to higher nitrogen in the leaf litter (Snowdon et al., 2005). Acacia are some of the dominant understorey species in the forests investigated in this study, and their presence may have

contributed to the high emissions of nitrogen-containing species; however, without fuel composition measurements, it is impossible to draw definitive conclusions.

P23, L1: Can you elaborate on how they would impact plume chemistry and influence air quality outcomes downwind of the fires? Some discussion would be helpful.

The sentence was modified to be more general (instead of focused on monoterpenes):

The initial mixture of trace gases emitted by a fire is one of the factors (along with meteorology and the presence of other sources) that influences plume aging (Akagi et al., 2012, Jaffe and Wigder, 2012) and therefore air quality outcomes downwind of the fires.

Technical Corrections:
P1, L17: Change "At a national level, average gross annual emissions of total carbon from fires.." to "..annual emissions of total carbon from some fires.." since not all vegetation grows back fast.

The other reviewer also had misgivings about this paragraph. We removed "rapid" and added a reference.

P3, L3: You already mention Hurst et al. 1996 in page 2, line 31. You should remove the sentence from the third page and add it to the second, or vice versa. Either way I think consolidating the statements would be helpful, since having it in two locations essentially saying the same thing seems redundant.

We removed the sentence from P2.

P3, L6: Abbreviate New South Wales National Parks as NSW. You mention it in Page 3, Line 11, but don't abbreviate it before then.

The abbreviation was added:

New South Wales (NSW) National Parks and Wildlife Service

P5, L5: "$CO_2$, CO, $CH_4$, acetic acid, ammonia, ethene … and "$CO_2$, CO, $CH_4$, ethane and ethene: : :" This could be considered "picky" but I think it would be useful to include the chemical formulas and names of all the compounds to maintain consistency (I.E: Carbon Dioxide ($CO_2$), Carbon monoxide (CO), Methane ($CH_4$), acetic Acid ($CH_3COOH$), ammonia ($NH_3$), etc).

Done.

P19, L17: "..emission factors listed in 5.." Do you mean Table 5?

Yes, thank you. The correction has been made.

Supplemental: No issues on the supplemental

References: Brilli, F., Gioli, B., Ciccioli, P., Zona, D., Loreto, F., Janssens, I. A., and

Ceulemans, R.: Proton Transfer Reaction Time-of-Flight Mass Spectrometric (PTRTOF-MS) determination of volatile organic compounds (VOCs) emitted from a biomass fire developed under stable nocturnal conditions, Atmospheric Environment, 97, 54-67, https://doi.org/10.1016/j.atmosenv.2014.08.007, 2014.

Liu, X., Huey, G. L., Yokelson, R. J., Selimovic, V., Simpson, I. J., Müller, M., Jimenez. J. L., Campuzano-Jost, P., Beyersdorf. A. J., Blake, D. R., Butterfield, Z., Choi, Y., Crounse, J. D., Day, D. A., Diskin, G. S., Dubey, M. K., Fortner, E., Hanisco, T. F., Hu, W., King, L. E., Kleinman, L., Meinardi, S., Mikoviny, T., Onasch, T. B., Palm, B. B., Peischl, J., Pollack, I. B., Ryerson, T. B., Sachse, G. W., Sedlacek, A. J., Shilling, J. E., Springston, S., St. Clair, J. M., Tanner, D. J, Peng, A. P., Wennberg, P. O., Wisthaler, A., and Wolfe, G. M.: Airborne measurements of western U.S wildfire emissions: Comparison with prescribed burning and air quality implications, J. Geophys. Res. Atmos., 122, 6108-6129, doi:10.1002/2016JD026315, 2017.

Koss, A. R., Sekimoto, K., Gilman, J. B., Selimovic, V., Coggon, M. M., Zarzana, K. J., Yuan, B., Lerner, B. M., Brown, S. S., Jimenez, J. L., Krechmer, J., Roberts, J. M., Warneke, C., Yokelson, R. J., and de Gouw, J.: Non-methane organic gas emissions from biomass burning: identification, quantification, and emission factors from PTRToF during the FIREX 2016 laboratory experiment, Atmos. Chem. Phys. Discuss., https://doi.org/10.5194/acp-2017-924, in review, 2017.

Knighton, W. B., Fortner, E. C., Midey, A. J., Viggiano, A. A., Herndon, S. C., Wood, E. C., and Kolb, C. E.: HCN detection with a proton transfer mass reaction spectrometer, International Journal of Mass Spectrometery, 283, 112-121, https://doi.org/10.1016/j.ijms.2009.02.013, 2009.

Selimovic, V., Yokelson, R. J., Warneke, C., Roberts, J. M., de Gouw, J., Reardon, J., and Griffith, D. W. T.: Aerosol optical properties and trace gas emissions by PAX and OP-FTIR for laboratory-simulated western US wildfires during FIREX, Atmos. Chem. Phys. Discuss., https://doi.org/10.5194/acp-2017-859, in review, 2017.

Yokelson, R. J., Burling, I. R., Urbanski, S. P., Atlas, E. L., Adachi, K., Buseck, P. R., Wiedinmyer, C., Akagi, S. K., Toohey, D. W., and Wold, C. E.: Trace gas and particle emissions from open biomass burning in Mexico, Atmos. Chem. Phys., 11, 6787-6808, https://doi.org/10.5194/acp-11-6787-2011, 2011.

[revised manuscript text omitted]

**S1    Additional information on prescribed fires**

As mentioned in the main text, we attended nine prescribed fires between 2010 and 2015. Seven of these fires were in the greater Sydney area in NSW, and two were in the State of Victoria. Table S1 lists the fires, their location, the dates on which they were sampled, the main vegetation type, the area burnt, the fuel loading, the time elapsed since the previous fire, the
5    coordinates of the sampling sites and the method(s) of sampling deployed. The number of grab samples collected at each fire is indicated in brackets in the last column of Table S1. For the NSW fires, the vegetation type, the area burnt, the fuel load and the time since last fire were sourced from the burn plans provided by the New South Wales National Parks and Wildlife Service. For the fires in Victoria, this information was gathered by the research team.

The emission factors from the open-path FTIR measurements at the Lane Cove, Turramurra, Abaroo Creek, Gulguer Plateau
10   and Alfords Point fires were reported in Paton-Walsh et al. (2014) but are reanalysed here to evaluate their dependence on modified combustion efficiency (MCE).

**S2    Details of the SIFT-MS analysis**

As described in the main text, the SIFT-MS was operated in multiple ion mode, targeting eighteen VOC species. The list includes aromatic species, nitrogen-containing species, some oxygenated species, some small hydrocarbons and some biogenic
15   species, targeting a breadth of chemical classes. Table S2 lists the species targeted, the reagent ion used, the mass-to-charge ratios measured and the calibration factors used to quantify them. It should be noted that hydrogen cyanide was assigned the same calibration factor as formaldehyde. Both species have a similar m/z (and are therefore likely to be transmitted in a similar way through the instrument), similar proton affinities, similar kinetics and little water dependence when measured by SIFT-MS (Španěl et al., 1999, 2004). Similarly, pyrrole was assigned the same calibration factor as isoprene. The instrument
20   response to monoterpenes was determined using $\alpha$-pinene and eucalyptol (1,8-cineole).

**Table S1.** Summary of prescribed fires in Australian temperate forest sampled in 2010-2013 and April 2015, including location, date, vegetation type, area burnt, pre-fire fuel loading, time elapsed since the area was last exposed to fire and sampling method(s) deployed. The number of grab samples collected at each fire is indicated in parentheses.

| Fire Name | Location | Date(s) | Vegetation | Area (ha) | Fuel load (t ha$^{-1}$) | Time since last fire | Lat, Lon of sampling site | Method(s) (# of samples) |
|---|---|---|---|---|---|---|---|---|
| Lane Cove | Lane Cove National Park, NSW | 31 Aug 2010 | Dry sclerophyll open woodland | 4.8 | 18-26 | unknown | -33.79, 151.15 | OP-FTIR[a] |
| Turramurra | Ku-Ring-Gai Chase National Park, NSW | 28 Sep 2010 | Dry sclerophyll shrubby forest/heath | 148.5 | 20-25 | unknown | -33.67, 151.15 | OP-FTIR[a] |
| Abaroo Creek | Heathcote National Park, NSW | 11&12 May 2012 | Dry sclerophyll shrubby forest/heath | 115 | 12.5 | 10 years | -34.10, 150.99 -34.13, 150.99 | Grab sampling (17) and OP-FTIR[a] |
| Gulguer Plateau | Gulguer Nature Reserve,NSW | 16 May 2012 | Dry scleropyll forest, grassy understorey | 32 | 8-10 | 30 years | -33.95, 150.62 | Grab sampling (9) and OP-FTIR[a] |
| Alfords Point | Georges River National Park, NSW | 23 May 2012 | Dry sclerophyll shrubby forest | 18 | 14-18 | 9 years | -33.99, 151.02 | Grab sampling (11) and OP-FTIR[a] |
| Prospect Reservoir | Prospect Nature Reserve, NSW | 27 Apr 2013 | Open woodland, grassy/shrubby understorey | 12.5 | 10-12 | >30 years | -33.81, 150.91 | Grab sampling (17) |
| Yeramba Lagoon | Georges River National Park, NSW | 26&27 Aug 2013 | Dry sclerophyll shrubby forest | 14 | 18 | unknown | -33.97,151.01 | Grab sampling (18) |
| Greendale | King Track, Greendale, VIC | 13 Apr 2015 | Heathy dry sclerophyll forest | 254 | 17 ± 2 | 32 years | -37.52,144.28 | OP-FTIR |
| Castlemaine | Kalimna Park, Castlemaine, VIC | 23 Apr 2015 | Heathy dry sclerophyll forest | 22 | 16 ± 2 | >30 years | -37.05, 144.24 | OP-FTIR |

[a] the emission factors from these OP-FTIR measurements were published in Paton-Walsh et al. (2014). The data are re-analysed to look at the dependence of emission factors on modified combustion efficiency (MCE) (see main text)

**Table S2.** Summary of SIFT-MS analysis of smoke samples: targeted species, selected masses, dwell time and sensitivity.

| Species Targeted | Reagent ion | m/z | Dwell time (ms) | Sensitivity (ncps ppb$^{-1}$) |
|---|---|---|---|---|
| $H_3O^+$ and clusters | $H_3O^+$ | 19, 37, 55 | 50 | – |
| $NO^+$ and clusters | $NO^+$ | 30, 48 | 50 | – |
| $O_2^+$ | $O_2^+$ | 32 | 50 | – |
| Acetaldehyde | $H_3O^+$ | 45 | 100 | 11.3 |
| Acetone | $H_3O^+$ | 59 | 100 | 14.1 |
| Acetonitrile | $H_3O^+$ | 42, 60 | 100 | 18.3 |
| Acetylene | $O_2^+$ | 26 | 100 | 4.4 |
| Benzene | $NO^+$ | 78 | 100 | 5.2 |
| 1,3-butadiene | $NO^+$ | 54 | 100 | 7.9 |
| Butanone | $NO^+$ | 102 | 100 | 11.4 |
| Ethanol | $NO^+$ | 45, 63 | 100 | 4.8 |
| Ethene | $O_2^+$ | 28 | 100 | 4.5 |
| Eucalyptol | $NO^+$ | 154 | 100 | 12 |
| Formaldehyde | $H_3O^+$ | 31 | 100 | 7.3 |
| Hydrogen cyanide | $H_3O^+$ | 28 | 100 | 7.3[a] |
| Isoprene (and furan) | $NO^+$ | 68 | 100 | 7.9 |
| Methacrolein (and methyl vinyl ketone) | $H_3O^+$ | 71 | 100 | 11.8 |
| Methanol | $H_3O^+$ | 33, 5 | 100 | 6.5 |
| Monoterpenes[b] | $H_3O^+$ | 81, 137 | 100 | 10.4 |
| Pyrrole | $H_3O^+$ | 68 | 100 | 7.9[c] |
| Toluene | $NO^+$ | 92 | 100 | 10.7 |
| Xylenes | $NO^+$ | 106 | 100 | 12 |

[a] assigned the same sensitivity as formaldehyde

[b] determined using $\alpha$-pinene and eucalyptol (1,8-cineole)

[c] assigned the same sensitivity as isoprene

**S3 Additional grab sampling results**

Emission ratios (ER) were derived for individual fires for all species measured by White cell FTIR and SIFT-MS in the grab samples. For some species at some fires, the correlations were poor ($R^2 < 0.5$) and these were excluded. Also, not every trace gas species was present at a detectable level in every sample. For some fires, this resulted in too few samples to allow an emission ratio to be meaningfully derived by regression for that species for a specific fire. Emission ratios for individual fires are listed in Table S3.

Figure S1 shows the correlation of ethane with CO for each of the five individual fires, and for all fires combined, as an example.

**S4 Additional open-path FTIR results**

All trace gases measured by open-path FTIR at the prescribed fires in Victoria exhibited strong correlations with either CO or $CO_2$. Correlations between the measured species at the Castlemaine fire are shown in Figure S2 as an example.

**Table S3.** Emission ratios determined at individual fires for species measured by SIFT-MS and White cell FTIR in grab samples of smoke

| Species | Ref. species | Abaroo Creek | $R^2$ | Alfords Point | $R^2$ | Gulguer Plateau | $R^2$ | Prospect Reservoir | $R^2$ | Yeramba Lagoon | $R^2$ | Mean (std. dev.) |
|---|---|---|---|---|---|---|---|---|---|---|---|---|
| **White cell FTIR** | | | | | | | | | | | | |
| CO | $CO_2$ | 0.15 ± 0.03 | 0.57 | 0.08 ± 0.02 | 0.62 | 0.44 ± 0.08 | 0.83 | 0.08 ± 0.02 | 0.89 | 0.18 ± 0.03 | 0.92 | 0.19 (0.15) |
| $CH_4$ | CO | 0.067 ± 0.009 | 0.86 | 0.065 ± 0.004 | 0.98 | 0.060 ± 0.009 | 0.79 | 0.037 ± 0.004 | 0.92 | 0.07 ± 0.01 | 0.89 | 0.06 (0.01) |
| Ethane | CO | 0.0045 ± 0.0007 | 0.83 | 0.0045 ± 0.0003 | 0.96 | 0.003 ± 0.001 | 0.76 | 0.0026 ± 0.0002 | 0.96 | 0.0055 ± 0.0006 | 0.97 | 0.004 (0.001) |
| **SIFT-MS** | | | | | | | | | | | | |
| Acetaldehyde | CO | 0.006 ± 0.004 | 0.99 | 0.0101 ± 0.0007 | 0.99 | 0.006 ± 0.002 | 0.63 | 0.010 ± 0.002 | 0.90 | 0.011 ± 0.005 | 0.96 | 0.009 (0.002) |
| Acetone | CO | 0.0034 ± 0.0009 | 0.85 | 0.0052 ± 0.0006 | 0.98 | 0.003 ± 0.001 | 0.80 | 0.0040 ± 0.0009 | 0.90 | 0.004 ± 0.003 | 0.90 | 0.0039 (0.0008) |
| Acetonitrile | CO | 0.0031 ± 0.0009 | 0.82 | 0.0050 ± 0.0005 | 0.98 | 0.0009 ± 0.0003 | 0.83 | 0.006 ± 0.002 | 0.94 | 0.005 ± 0.001 | 0.98 | 0.005 (0.001) |
| Benzene | Ethene | 0.09 ± 0.02 | 0.64 | 0.068 ± 0.004 | 0.98 | 0.10 ± 0.02 | 0.58 | 0.088 ± 0.002 | 0.99 | 0.07 ± 0.01 | 0.99 | 0.08 (0.01) |
| Butadiene | Ethene | 0.048 ± 0.003 | 0.93 | 0.047 ± 0.003 | 0.97 | 0.037 ± 0.005 | 0.82 | 0.04 ± 0.01 | 0.95 | 0.045 ± 0.005 | 0.96 | 0.042 (0.006) |
| Ethanol[b] | CO | | | 0.00021 ± 0.00005 | 0.97 | | | | | | | |
| Furan + isoprene | CO | 0.0022 ± 0.0003 | 0.83 | 0.0018 ± 0.0002 | 0.96 | 0.0023 ± 0.0009 | 0.75 | 0.0009 ± 0.0005 | 0.65 | 0.0017 ± 0.0002 | 0.85 | 0.0018 (0.0006) |
| Methanol | CO | 0.029 ± 0.004 | 0.88 | 0.028 ± 0.003 | 0.95 | | | 0.016 ± 0.004 | 0.52 | 0.027 ± 0.009 | 0.66 | 0.025 (0.006) |
| Toluene | CO | 0.0004 ± 0.0002 | 0.81 | 0.00086 ± 0.00003 | 0.98 | 0.0004 ± 0.0003 | 0.63 | 0.00045 ± 0.00009 | 0.63 | 0.0007 ± 0.0004 | 0.89 | 0.0006 (0.0002) |
| mean MCE of samples | | 0.89 ± 0.05 | | 0.93 ± 0.02 | | 0.78 ± 0.09 | | 0.92 ± 0.03 | | 0.89 ± 0.06 | | 0.88 (0.07) |

[Figure]

**Figure S1.** Emission ratio of ethane to CO for each individual fire sampled by grab sampling and for all the fires combined.

[Figure]

**Figure S2.** Correlation plots for open-path FTIR measurements at the Castlemaine, VIC fire.